# Land conversion to agriculture induces taxonomic homogenization of soil microbial communities globally

Ziheng Peng [1], Xun Qian[2], Yu Liu[1], Xiaomeng Li[1], Hang Gao[1], Yining An[1], Jiejun Qi[1], Lan Jiang[2], Yiran Zhang[2], Shi Chen[1], Haibo Pan[1], Beibei Chen[1], Chunling Liang[1], Marcel G. A. van der Heijden [3,4], Gehong Wei[1] ✉ & Shuo Jiao [1] ✉

Agriculture contributes to a decline in local species diversity and to above- and below-ground biotic homogenization. Here, we conduct a continental survey using 1185 soil samples and compare microbial communities from natural ecosystems (forest, grassland, and wetland) with converted agricultural land. We combine our continental survey results with a global meta-analysis of available sequencing data that cover more than 2400 samples across six continents. Our combined results demonstrate that land conversion to agricultural land results in taxonomic and functional homogenization of soil bacteria, mainly driven by the increase in the geographic ranges of taxa in croplands. We find that 20% of phylotypes are decreased and 23% are increased by land conversion, with croplands enriched in *Chloroflexi, Gemmatimonadota, Planctomycetota, Myxcoccota* and *Latescibacterota*. Although there is no significant difference in functional composition between natural ecosystems and agricultural land, functional genes involved in nitrogen fixation, phosphorus mineralization and transportation are depleted in cropland. Our results provide a global insight into the consequences of land-use change on soil microbial taxonomic and functional diversity.

Due to increasing human activities and agricultural intensification, an emerging body of research suggests that ecological communities are undergoing fundamental changes across various spatial dimensions[1]. Most studies investigating the consequences of land-use changes and agricultural expansion on ecological communities have focused on local species diversity[2–4] due to its ease of measurement and monitoring[5]. Such studies are relevant to highlight the loss of global biodiversity loss and species extinction[6–8]. However, in addition to reducing local species diversity, agricultural conversion also caused biotic homogenization at larger spatial scales[9–11], posing a significant concern for ecosystem services and conservation.

Biotic homogenization refers to the increase in taxonomic or functional similarities among ecological communities distributed spatially over time[12]. Biotic homogenization can be quantified by a decrease in β-diversity, e.g., a decrease in compositional dissimilarity between sites. Biotic homogenization can occur due to the establishment of exotic species (increasing similarity between communities), the loss of native species specific for a limited set of locations (reducing similarity) or most likely a combination of both[13,14]. Indeed, both

[1]State Key Laboratory for Crop Stress Resistance and High-Efficiency Production, Shaanxi Key Laboratory of Agricultural and Environmental Microbiology, College of Life Sciences, Northwest A&F University, 712100 Yangling, Shaanxi, P. R. China. [2]College of Natural Resources and Environment, Northwest A&F University, 712100 Yangling, Shaanxi, P. R. China. [3]Plant–Soil Interactions Group, Agroscope, Zurich, Switzerland. [4]Department of Plant and Microbial Biology, University of Zurich, Zurich, Switzerland. ✉e-mail: weigehong@nwsuaf.edu.cn; shuojiao@nwsuaf.edu.cn

natural pressures and anthropogenic activities, such as climate change, agricultural expansion, urbanization and habitat homogenization, could cause biotic homogenization[9,15–19]. So far, the impact of land use and agricultural conversion on biotic homogenization mainly focused on aboveground habitats[18], with limited attention given to belowground communities. The information about agriculture-induced biotic homogenization of belowground communities is essential for regional biodiversity planning and conservation purposes.

Land-use change and agricultural conversion can alter community assembly processes, community composition and species diversity concurrently[4,20–22]. These changes are underpinned by species extinction, colonization and uneven shifts in relative abundance among different geographic regions. Intense agriculture can contribute to soil compaction, salinization, acidification, metal accumulation, organic matter loss and nutrient imbalance[23]. These related environmental stressors generally induce structural shifts in microbial taxonomic and functional composition[24,25], such as the retention of acid-tolerant taxa and the loss of specific functional traits for pathogen suppression or crop fitness[26,27]. Consequently, these shifts create ecological feedbacks that further influences soil functions critical for maintaining soil health and agricultural productivity. Despite numerous studies examining the responses of microbiome composition and function to agricultural conversion[28–31], these observations are predominantly site-specific and limited to a local scale[3], making it challenging to infer whether shifts in specific microbial taxa are relevant to the diverse range of soils worldwide[32]. Currently, we still lack a generalizable and consistent understanding of how soil microbial taxonomic and functional profiles respond to agricultural conversion and which microbial lineages and functions are mostly impacted across a wide range of soil and climate types. This knowledge gap hinders our comprehensive understanding of the global decline in biodiversity and associated ecosystem functions.

In the present study, we address two major questions: (1) whether agricultural effects lead to taxonomic and functional biotic homogenization of soil microbiomes at large spatial scales? (2) how land-use changes alter soil microbial community composition and functions across a wide range of soil and climate types, and which microbial lineages and functions most strongly impacted? We combined a continental soil survey and a global-scale meta-analysis to address these questions. For the first question, we conducted a continental soil survey of 1185 samples from agricultural fields and the adjacent natural ecosystems (covering forest, grassland, wetland; Fig. 1c) across China to provide large-scale evidence of agriculture-induced taxonomic and functional homogenization of soil microbiomes. To gain a global perspective on agricultural-induced biotic homogenization, and to complement the continental scale soil survey, we also collected 16S rRNA amplicon-based sequencing data from soil samples of global agricultural-natural ecosystem pairs from all available gene banks (Fig. 1a). We hypothesized that agricultural conversion causes taxonomic and functional homogenization of soil microbiomes. For the second question, we used the continental survey dataset to explore general patterns of soil microbiome taxonomic and functional responses to agricultural conversion across a wide range of soil and climate types. We also determined how these responses vary among ecosystem types and different microbial lineages. Our results demonstrate that land-use change for agricultural purposes reduces taxonomic diversity in soil bacterial communities.

## Results

### Agriculture causes biotic homogenization of taxonomic and functional profiles
Our continental soil survey dataset (Fig. 1c and Supplementary Tables 1, 2) revealed that $\beta$-diversity of both microbial taxonomic and functional composition (identified by KEGG and COG) was significantly lower in cropland than in natural soil (Fig. 1d, h and Supplementary Figs. 1b, 2) demonstrating that cropland soils are more similar than paired natural ecosystem soils. For example, the $\beta$-diversity of taxonomic composition was significantly lower in cropland than in forest ($F_{1,91504} = 6.429$, slope = 0.0016, $p < 0.05$), in grassland ($F_{1,75348} = 1532$, slope = 0.0276, $p < 0.001$), and in wetland ($F_{1,77004} = 6450$, slope = 0.0532, $p < 0.001$; Fig. 1d). The $\beta$-diversity of functional composition was significantly lower in cropland than in grassland ($F_{1,88} = 9.021$, slope = 0.0885, $p < 0.01$), and in wetland ($F_{1,88} = 6.886$, slope = 0.0527, $p < 0.05$; Fig. 1h). When considering $\beta$-diversity at the site-level and keeping the same number of samples across sites, significant lower value was also detected in croplands compared to forest, grassland and wetland (Supplementary Fig. 3). The global meta-analysis based on more than 2400 soil samples across six continents (Fig. 1a) further showed that microbial communities from croplands were significantly different from forest soils in taxonomic composition (PERMANOVA; $R^2 = 0.026$, $p < 0.001$), and $\beta$-diversity was significantly lower in croplands than forest soils at global scale (wilcoxon test: $p < 0.001$; Fig. 1b). These results jointly provide large-scale, e.g., continental and global scale, evidence for biotic homogenization of soil microbiome under agricultural conversion.

Moreover, we found that the phylotypes, that are present in both croplands and natural ecosystems, were found in significantly more samples of croplands than in natural ecosystems (wilcoxon test: $p < 0.001$; Supplementary Fig. 4a), indicating an increase in the geographic ranges of existing taxa in croplands. The phylotypes unique to natural ecosystems occurred in significantly fewer samples than other shared phylotypes that present in both croplands and natural ecosystems (wilcoxon test: $p < 0.001$; Supplementary Fig. 4b), implying a possible loss of these habitat-specific taxa after agricultural conversion. Given that microbial composition is critical for maintenance and resilience of soil functions, e.g., nutrient supply, litter decomposition and water regulation[33,34], agricultural-induced biotic homogenization could cause ecosystem service degradation and threaten sustainable management. Thus, even though a few studies have assessed the impacts of agricultural land-use change on microbial diversity and composition, biotic homogenization along with the reduction of regional community heterogeneity at large spatial scales should be taken into full consideration as a significant consequence of agricultural conversion.

### Agricultural effects on specific bacterial phylotypes and functions
To provide a general insight into agricultural-induced shifts in microbial phylotypes and functions originated from multiple natural ecosystems, we compared differences in community structure between cropland and surrounding forests, grasslands, and wetlands, respectively. Microbial communities of cropland soils were significantly different from those of natural forest, grassland and wetland soils in taxonomic composition (Fig. 1e–g and Supplementary Fig. 5). These differences were evident at both the phylum, class, order (Supplementary Fig. 5) and phylotypes levels (Fig. 1e–g). Notably, the largest differences in taxonomic composition were found between croplands and wetlands (PERMANOVA; $R^2 = 0.060$, $p < 0.001$), followed by the comparison of croplands and forests (PERMANOVA; $R^2 = 0.052$, $p < 0.001$), and between croplands and grasslands (PERMANOVA; $R^2 = 0.038$, $p < 0.001$). Specifically, agricultural impacts significantly altered microbial composition (PERMANOVA; $p < 0.05$) in almost all of the locations, except for 2 of 37 sites in croplands and grasslands (Supplementary Table 3). On average, agricultural effects significantly altered the abundance of nearly half of the phylotypes (44% for forests, 41% for grasslands, and 45% for wetlands; Supplementary Fig. 6a, b). Approximately 20% of the ASVs were lost from natural ecosystems upon conversion to agriculture, while approximately 23% of the ASVs increase in abundance. Specifically, the relative abundance of

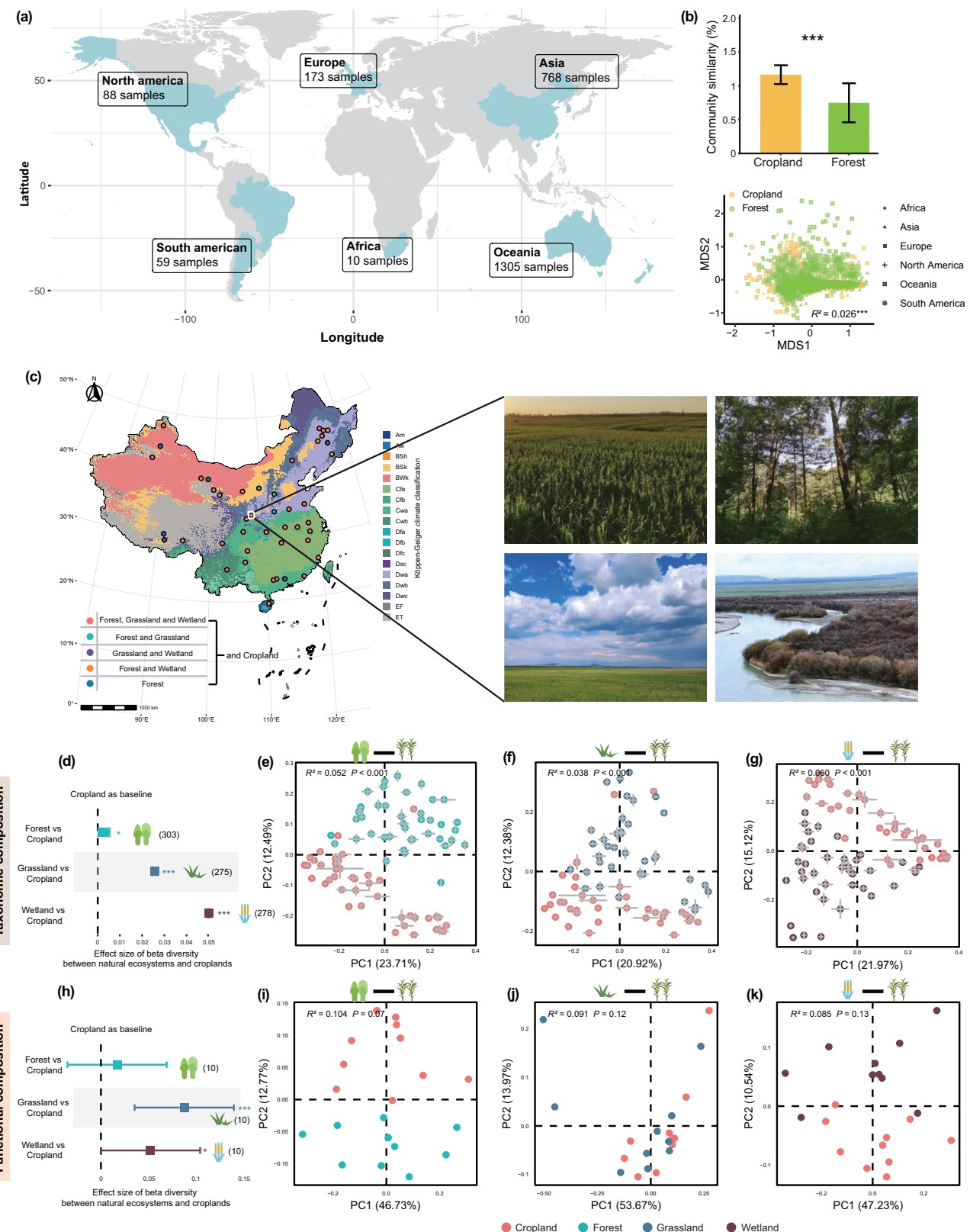

*Chloroflexi, Gemmatimonadota, Planctomycetota, Myxcoccota* and *Latescibacterota* increased in croplands compared with all three natural ecosystems (Fig. 2 and Supplementary Fig. 7), indicating that these taxa exhibited consistent responses to agricultural conversion across a broad range of habitat types. In addition, changes in the abundance of dominant phylotypes were mainly related to soil pH and moisture between ecosystems (Fig. 2b, c and Supplementary Fig. 7c, f).

Interestingly, the effects of agricultural conversion were much lower when focusing on the functional composition identified by KEGG (Fig. 1i–k) and there were no significant differences between agricultural and natural ecosystems. Moreover, the functional composition identified using COGs exhibited significant but minor differences between cropland and natural ecosystems (Supplementary Fig. 1c). Only less than 10% of functional groups identified by KEGG and COGs

**Fig. 1 | Taxonomic and functional homogenization of microbial communities in response to agricultural impacts at global and continental scale. a** Data distribution of one-to-one correspondence of 2403 sequencing data between cropland and natural ecosystems across countries and continents. **b** Response of community similarity to agricultural conversion. The each bars represent the mean ± standard errors (SE). Asterisks indicate significant difference (wilcoxon test, ***$p < 0.001$). Non-metric multidimensional scaling of Bray–Curtis distances showing community dissimilarities between cropland and forest. Cropland, 1033 samples; Forest, 1370 samples. **c** Map showing 44 regions covering croplands and adjacent natural ecosystems. Typical terrestrial ecosystems, including croplands, forests, grasslands and wetlands. **d, h** Effect sizes of natural ecosystems impacts on $\beta$-diversity in taxonomic composition (**d**) and functional composition annotated with KEGG (**h**) relative to croplands. The estimated effect sizes are regression coefficients based on the linear models. Data are presented as mean ±

s.e.m. of the estimated effect sizes. Sample size is showed by number of data pairs for each group. Statistical significance is based on $F$-test; ***$p < 0.001$, **$p < 0.01$, *$p < 0.05$. **e–g** Principal coordinate analyses of Bray–Curtis distances showing dissimilarities among taxonomic composition between croplands and natural ecosystems, including forests (**e**), grasslands (**f**), and wetlands (**g**). A total of 303 forest-cropland pairs (**e**), 275 grassland-cropland pairs (**f**), and 278 wetland-cropland pairs (**g**) were compared. Each point indicates a site, and error bars around the means represent standard error of samples in given a site. **i–k** Principal coordinate analyses of Bray–Curtis distances showing dissimilarities among functional composition annotated with KEGG between croplands and natural ecosystems, including forests (**i**), grasslands (**j**), and wetlands (**k**). A total of 10 forest-cropland pairs (**i**), 10 grassland-cropland pairs (**j**), and 10 wetland-cropland pairs (**k**) were compared. Communities differed among ecosystem types using PERMANOVA: ***$p < 0.001$, **$p < 0.01$, *$p < 0.05$.

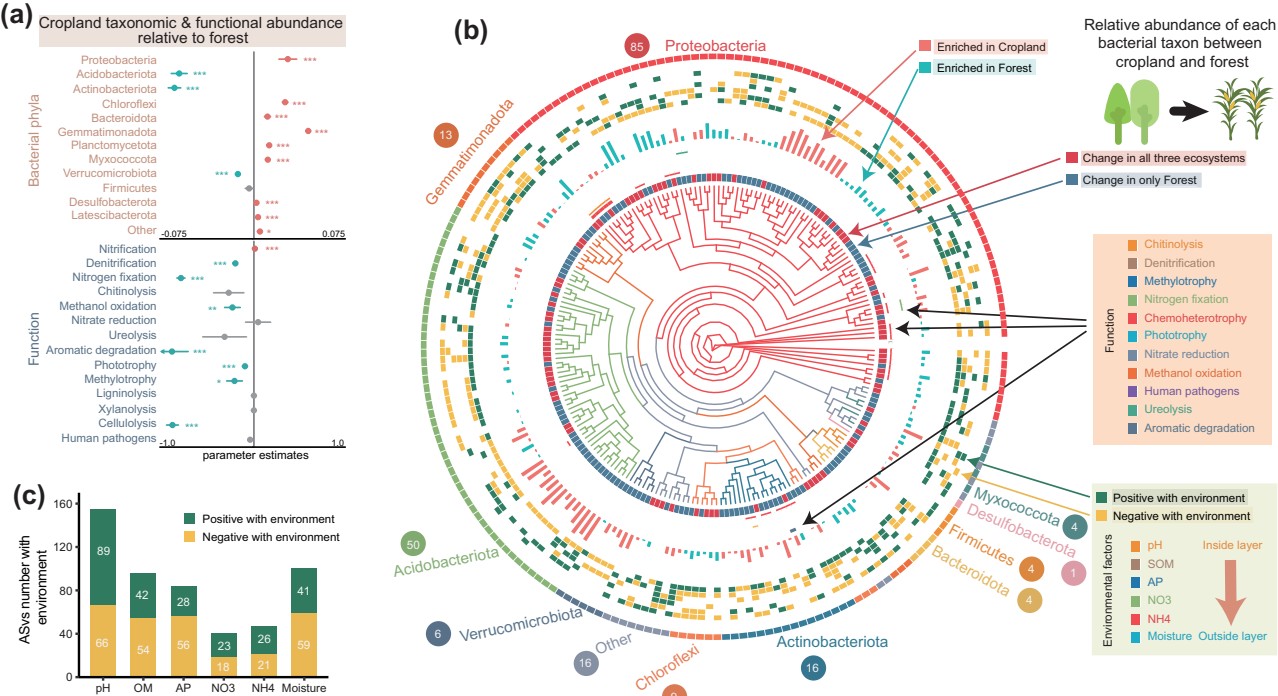

**Fig. 2 | Effects of agricultural conversion on different microbial taxa compared with forests at continental scale. a** Effect sizes of agricultural impacts on the relative abundance of major microbial taxonomic groups and functional groups as classified by FAPROTAX compared with forests. A total of 303 forest-cropland pairs were compared. The estimated effect sizes are regression coefficients based on the linear mixed-effects models. Data are presented as mean ± s.e.m. of the estimated effect sizes. Statistical significance is based on Wald type II $\chi^2$ tests; ***$p < 0.001$, **$p < 0.01$, *$p < 0.05$. Non-significant changes are denoted by gray dots. **b** The phylogenetic relationships of individual microbial phylotypes with a significant response ($p < 0.05$) based on Wald test using DESeq2 to agricultural impacts and with average relative abundance > 0.05% among croplands and forests. Colors of the branches in the first and sixth rings correspond to individual phyla. Colors of

the second ring represent phylotypes with significant increase or decrease under agricultural impacts. Colored blocks of the third ring represent the performance of a particular function by at least one phylotypes. The bars of the fourth ring represent the positive and negative effect sizes of agricultural impacts on relative abundances of phylotypes. Colored blocks of the fifth ring represent Spearman's correlation between the relative abundance of phylotypes and soil physicochemical properties (from inner to outer rings: pH, SOM, AP, $NH_4$, $NO_3$, and moisture). **c** Bar plots show the number of phylotypes that were significantly correlated with specific soil physicochemical properties. pH, soil pH; SOM, soil organic matter; AP, available phosphorus; $NH_4$, ammonium nitrogen; $NO_3$, nitrate nitrogen. Source data are provided as a Source Data file.

were affected by agricultural conversion (10%, 3% and 8% of KOs, and 5%, 1%, and 15% of COGs when comparing cropland with forest, grassland and wetland, respectively; Supplementary Fig. 6c-f). In terms of functional composition, agriculture significantly decreased the abundance of bacterial taxa specialized in nutrient cycling (for example, nitrogen fixation, phototrophy, and aromatic degradation) as classified by FAPROTAX compared with natural ecosystems (Fig. 2a and Supplementary Fig. 7a, c)[35]. Specific functional shifts were also observed in the metagenomic dataset (Fig. 3 and Supplementary Fig. 8). In total, three categories showed a consistent change in

direction compared with other three natural ecosystems when aggregating over level 3 functional categories through COG annotations (Supplementary Fig. 8). The functional categories "translation, ribosomal structure and biogenesis", and "cytoskeleton" increased while "defense mechanisms" diminished in croplands. However, specific carbon-degrading genes exhibited inconsistent effects upon agricultural conversion (some genes were enriched or deleted), while significant differences in the overall carbon metabolism were not detected under agricultural land-use (Fig. 3a, b). This is most likely due to the high redundancy of broadly distributed functions, thereby

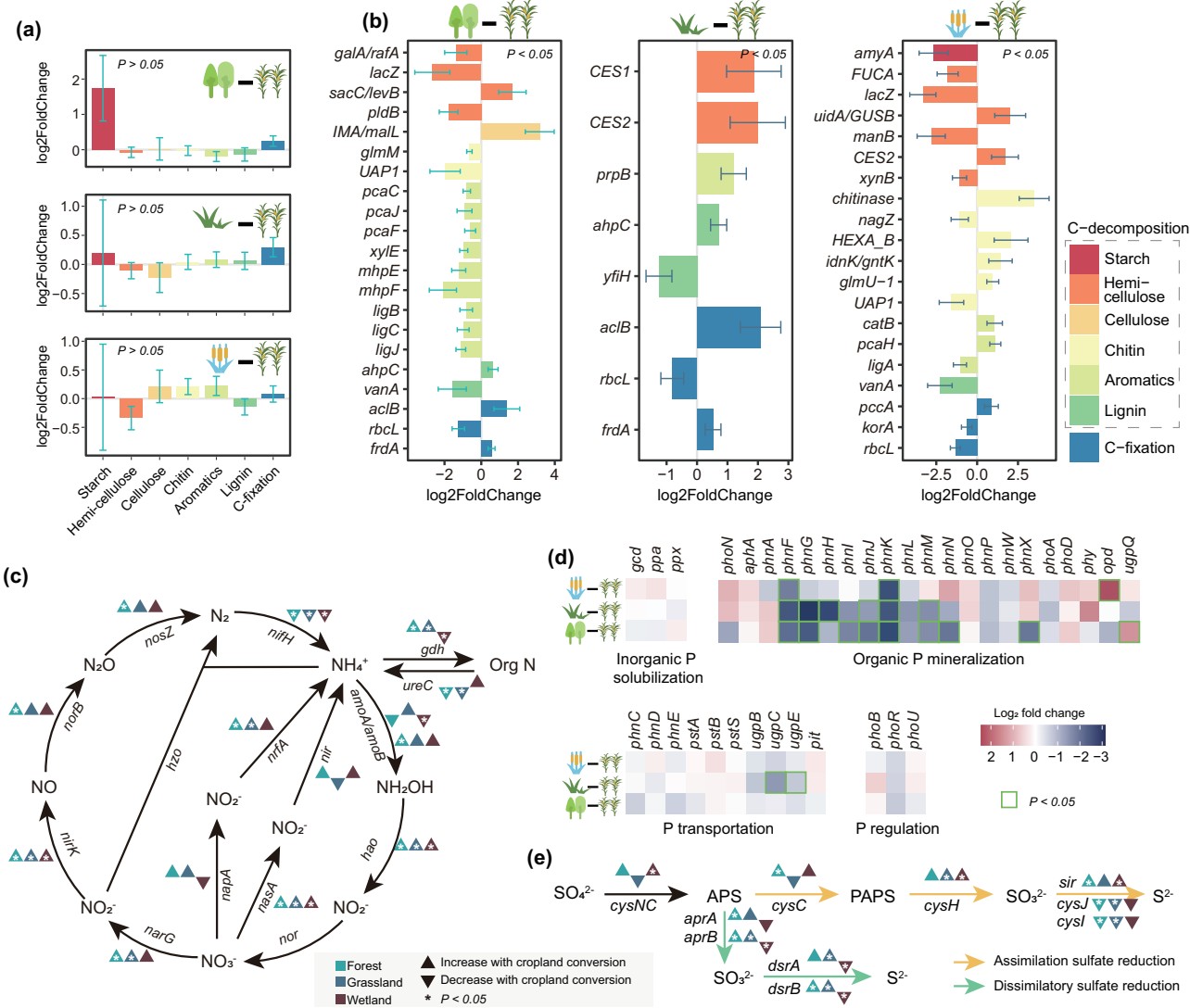

**Fig. 3 | Agricultural effects on functional genes involved in biogeochemical cycling processes at continental scale. a** C degradation and fixation from the metagenomic shotgun sequence. Bars represent log2-fold changes (LFC) in functional genes involved in C degradation and fixation aggregated over major substrates in croplands relative to forests, grasslands and wetlands. **b** C degradation and fixation from the metagenomic shotgun sequence. Specific functional genes involved with C degradation and fixation with a significant LFC with padj. <0.05 are shown based on Wald test using DESeq2. **c** N processes from the metagenomic shotgun sequence. Genes where LFC was significant ($p < 0.05$) are labeled in asterisks. **d** Phosphorus cycling from metagenomic shotgun sequence. Heatmaps represent the log2 fold-difference calculated for each gene. Significant differences between croplands and natural ecosystems are highlighted with green squares. **e** Sulfur cycling from metagenomic shotgun sequence. Yellow and green arrows represent assimilatory and dissimilatory nitrate and sulfate reduction, respectively. Genes where LFC was significant ($p < 0.05$) are labeled in asterisks. Each bars represent the LFC ± s.e.m. $N = 10$ biologically independent samples for each ecosystems. Source data are provided as a Source Data file.

buffering against taxonomic changes induced by agricultural land-use. Indeed, broad functions such as respiration, overall carbon catabolism and anabolism often seem more stable to shifts in microbial taxonomic composition than narrow metabolic functions such as the degradation of specific substrate[36–38].

Agriculture significantly altered a number of functionally important for N cycling, P utilization and sulfur metabolism genes. First, agriculture appeared to increase nitrification and denitrification processes, as indicated by increased *nirK*, *narG*, *amoB* and *hao* genes and it decreased the abundance of nitrogen fixation (*nifH*) (Fig. 3c), which could be due to the application of fertilizers and/or the loss of leguminous plant taxa found in natural ecosystems[39]. These results are in agreement with the increase of $N_2O$ production and the decrease of nitrogen fixation upon land-use change[40,41]. The abundance of key genes for organic P mineralization and transportation (for example, *phn* and *ugp*) were decreased in cropland (Fig. 3d). Opposite to this, the

dissimilatory sulfate reduction genes (*apr* and *dsr*) had higher abundance in croplands than in forests and grasslands but lower than in wetlands (Fig. 3e).

## Mechanisms underlying changed bacterial communities

A set of specific microbial traits associated with microbial dormancy and dispersal would regulate their ability to survive in land-use change associated with resource-based and disturbance-based scenarios[42]. For example, the abundance of Firmicutes and Actinobacteria with spore-forming ability was lower in croplands compared to three other natural ecosystems (see Fig. 2), which was closely linked to a decrease in community-aggregated dormancy strategies (Supplementary Fig. 9). We also observed that resuscitation-promoting gene was increased in cropland (Supplementary Fig. 9), which are associated with long-term persistence of viable bacterial populations[43], indicating that the resuscitation after disturbance can

allow for the proliferation of dormant taxa and accelerate increases in species richness[42].

Moreover, homogeneous selection (HoS; selection under homogeneous abiotic and biotic conditions in space and time) dominated microbial community assembly (as calculated using $\beta$NTI ($\beta$-nearest taxon index) and Raup–Crick based on Bray–Curtis dissimilarity (RC$_{bray}$) analysis) in croplands, with relative importance of 94.6% (Supplementary Fig. 9). At the same time, agriculture, acting as an environmental filter, continues to enhance homogeneous selection on microbial assembly processes (Fig. 4b), as crop management result in homogeneous abiotic and biotic conditions across space. Our results suggest that both microbial traits and environmental filtering could play prominent roles in regulating agricultural-induced microbial composition shifts.

Biotic interactions and abiotic environmental conditions also affect microbial composition under land-use change (Fig. 4a). Taxonomic composition showed significant correlations with environmental filtering of soil pH, moisture, and NH$_4$-N content, the heterogeneity of soil pH and NH$_4$-N content, and soil saprotrophic and pathogenic fungi. Functional composition was highly correlated with environmental filtering and heterogeneity of soil pH and NH$_4$-N content. To disentangle direct and indirect impacts of land-use change and environmental drivers on microbial composition, we performed structural equation modeling (SEM; Supplementary Fig. 11) using the most important soil and biotic explanatory variables, such as saprotrophic and pathogenic fungi, which were not collinear among them. Fungal saprotrophic and pathogenic composition, which was also affected by agricultural land-use, were significantly and directly

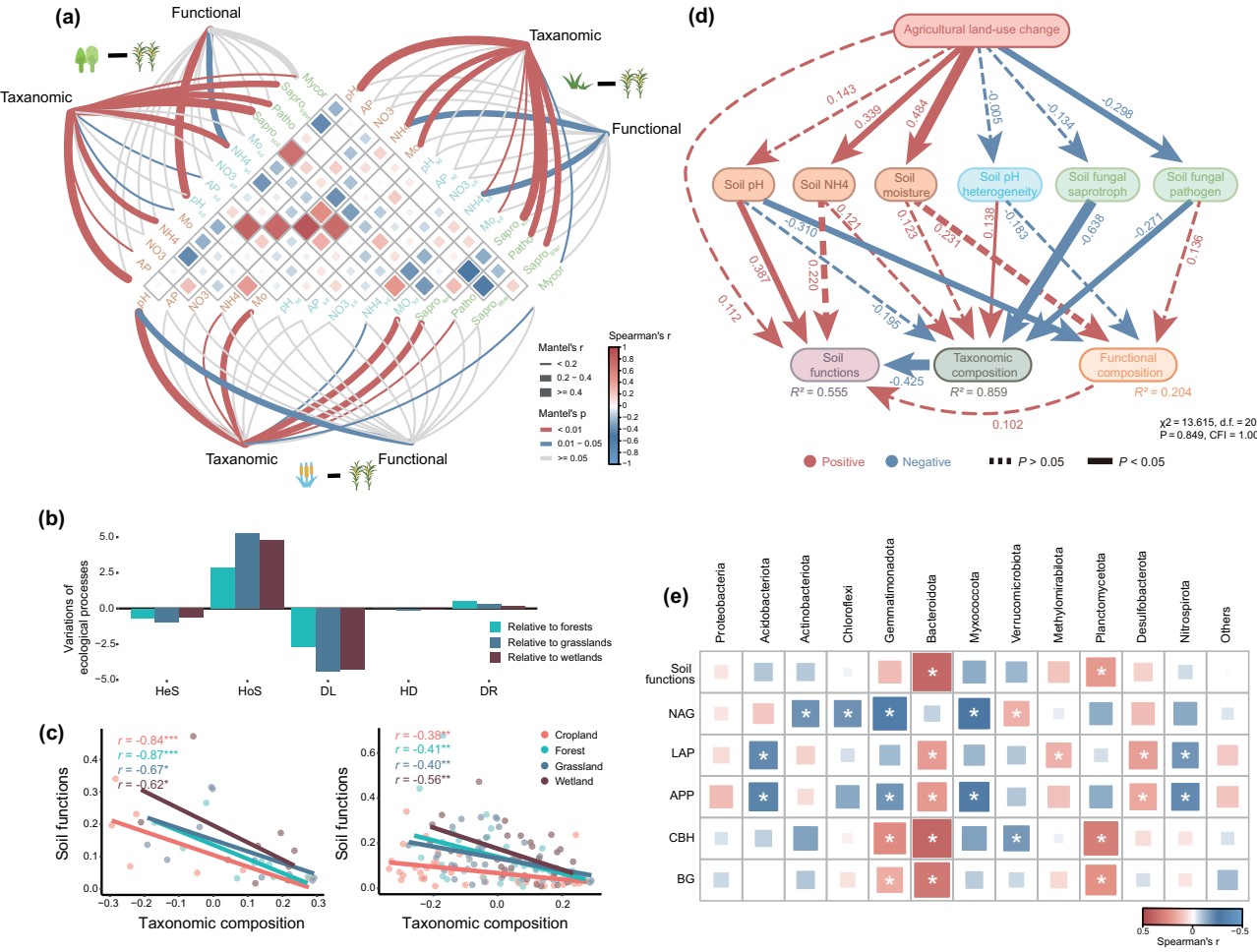

**Fig. 4 | Environmental drivers of microbial composition and their relationship to soil functions at continental scale. a** Pairwise comparisons of environmental factors are shown, with a color gradient denoting Spearman's correlation coefficients. Taxonomic and functional (based on KEGG modules) community composition was related to each environmental factor by Mantel tests. Edge width corresponds to Mantel's $r$ statistic for the corresponding distance correlations, and edge color denotes the statistical significance based on 9999 permutations. A total of 303 forest-cropland pairs, 275 grassland-cropland pairs, and 278 wetland-cropland pairs were tested by Mantel analysis. **b** Agricultural-induced change of relative importance of community assembly processes from forests, grasslands and wetlands. **c** The strong correlations between taxonomic composition (the first PC axis) and soil functions were assessed using Spearman's correlation based metagenomic data sites (Left) and all sites (Right). **d** Structural equation models (SEMs) showing the relationships among agricultural impacts, soil and fungal variables, microbial taxonomic and functional composition, and soil functions. Red and blue arrows indicate positive and negative relationships, respectively. Solid or dashed

lines indicate significant ($p < 0.05$) or non-significant relationships. Numbers near the pathway arrow indicate the standard path coefficients. Comparative fit index (CFI) = 1.00, and $n = 160$ sites of four ecosystems. R$^2$ represents the proportion of variance explained for every dependent variable. **e** Heatmap showing the correlations between the relative abundance of major microbial taxonomic groups and soil functions. The color denotes the correlation coefficient determined by Spearman's correlation. Asterisk indicate a significant correlation ($P < 0.05$). pH, soil pH; AP, available phosphorus; NH4, ammonium nitrogen; NO$_3$, nitrate nitrogen. Mo, soil moisture; Sapro$_{soil}$, soil fungal saprotrophs; Sapro$_{litter}$, litter fungal saprotrophs; Patho, plant fungal pathogens; pH$_{sd}$, standard deviation of soil pH; AP$_{sd}$, standard deviation of soil AP; NH4$_{sd}$, standard deviation of soil NH4; NO3$_{sd}$, standard deviation of soil NO3; Mo$_{sd}$, standard deviation of soil moisture; BG, $\beta$–1,4-glucosidase; CBH, 1,4-$\beta$-Dcellobiohydrolase; BX, $\beta$-xylosidase; NAG, $\beta$–1,4-N-acetylglucosaminidase; LAP, L-leucine aminopeptidase; APP, alkaline phosphatase. Source data are provided as a Source Data file.

correlated with bacterial taxonomic composition (Fig. 4a, d). Soil pH filtering played the strongest role in shaping taxonomic and functional composition (Fig. 4a, d). Moreover, the association of fungal and bacterial communities suggest an important role for biotic interactions in mediating agricultural-induced microbial composition changes. Although these variables could explain 85% of the variations in taxonomic composition, only 20% of the variations in functional composition were explained due to functional redundancy. More in-depth studies are necessary to determine the main drivers of changes in microbial functional composition.

### Links between bacterial communities and soil functions

Soil enzyme functions involved in carbon, nitrogen, and phosphorus cycling differed between cropland and natural ecosystems (Fig. 4c, d). Interestingly, we did not observe the relationship between microbial functional composition and soil enzyme functions. We also found that the association of soil enzyme functions with microbial composition varied among different microbial lineages (Fig. 4e). The relative abundance of *Bacteroidota* was positively correlated with soil functions and activities of four of the five enzymes while the relative abundance of *Gemmatimonadota* were positively correlated with $\beta-1,4$-glucosidase (BG) and $\beta$-D-cellobiosidase (CBH) and negatively correlated with $\beta-1,4$-acetylglucosaminidase (NAG) and alkaline phosphatase (APP). In all, these results indicate significant linkages between soil functions and microbial taxonomic composition but not functional composition.

## Discussion

Agricultural land-use change has exerted profound effects on above- and belowground biodiversity[2,44], and the effects are likely to accelerate in the coming decades[18]. While a number of studies showed that agricultural conversion led to biotic homogenization of aboveground communities, still very few studies investigated the belowground consequences. In the present study, we summarized the generalized effects of land-use conversion on belowground microbial communities and functions, encompassing multiple ecosystems. Our study provides large-scale evidence of taxonomic and, to a lesser degree, functional homogenization of soil microbiomes following agricultural conversion in terrestrial ecosystems at global and continental scales. The taxonomic variation across sites (Beta-diversity) was significantly lower in croplands than in grasslands, wetlands, and forests, pointing to biotic homogenization in croplands.

Although land-use changes and agricultural conversion have been proven to be major drivers of biodiversity loss[45,46], positive impacts of agriculture on biodiversity have been observed at regional and local scales in some studies[47–49]. One facet of these trends is that although local or alpha diversity may increase, this is typically at the expense of beta diversity[12]. Previous studies have demonstrated that increases in local land use intensity led to biotic homogenization of microbial, plant, and animal groups both above- and below-ground[4,25]. Biotic homogenization is largely independent of changes in alpha diversity; land use intensity reduced local alpha-diversity in aboveground groups, but increased the α-diversity in belowground groups[4]. Our study further extends these earlier observations at a continental and global scale and now provides widespread evidence that agricultural conversion results in biotic homogenization of the soil microbiome. Although taxonomic homogenization in cropland versus natural ecosystems was stronger and more significant in many cases, we observed very important microbial functional shifts under croplands, including functional homogenization[15,50]. This was evident when we calculated the beta-diversity across sites based on functional gene composition. Since the functional components of biodiversity are fundamental parts of ecosystem functions and services[51,52], functional homogenization is the most direct evidence for the potential loss of ecosystem functions

caused by agricultural conversion[16,53]. Our findings extend taxonomic-level results in Amazonian Forest[21] and European grasslands[4] that focus on the impact of agricultural management on belowground taxonomic homogenization in local-scale, to the large-scale functional homogenization. Overall, our study provides a comprehensive insight that agricultural land-use change cause biotic homogenization in taxonomic and functional composition, and suggests halting reclamation and developing ecological restoration for cropland to conserve landscape-scale biodiversity and ecosystem service provision[53,54].

Biotic homogenization in response to agricultural impacts is a multifaceted process that involves considering the invasion and extinction of species, as well as the heterogeneity of landscapes. In agricultural systems, it is generally believed that the biomes are a subset of the regional species pool, which is composed of surrounding natural ecosystems[55]. This highlights the selective effects of agricultural conversion, which could cause pressure and force on soil communities from natural ecosystems. For example, the destruction of soil structure and aggregates, as well as alterations and homogenization in soil environmental conditions caused by agricultural conversion can result in the trait-based filtering out of certain species, leading to the loss of existing species and the dominance of microorganisms that are better adapted to agricultural management. Moreover, geographic range size is a major determinant of species' extinction risk, and rare species therefore are vulnerable to land use change and are at greater risk of extinction[25]. The establishment of agricultural systems through intensive management can facilitate the spread of colonizing species that are abundant and prevalent due to the characteristics of broad environmental adaptation, while rare or specialized species may decrease in their abundance and occupancy over time[56,57], which led to a homogenization of community composition across space. Land-use change is proposed to affect turnover in community composition via its effect on stress tolerance, resource acquisition, and dispersal ability. Stronger stress-tolerant, broader resource-flexibility cosmopolitan species with unlimited dispersal capacity are more stable to land-use change because of increasing adaptive potential and/or extensive ability to exploit soil resource availability[58,59]. Frequently disturbed soil environments can promote the gains and proliferation of novel species and the gradual replacement of locally distinct communities by cosmopolitan communities via altered competitive and coexistence dynamics[60], homogenizing assemblage composition.

On the other hand, the influence of agricultural conversion on biotic homogenization might be attributed to the reduction in environmental heterogeneity in monoculture-dominated landscapes[61]. Landscape heterogeneity is central to the spatial organization of ecological communities[62]. Variations in vegetation structural and soil conditions influence beta diversity and turnover of soil fauna, bacteria, and fungi. Monoculture-dominated croplands have lower environmental heterogeneity compared with vegetation structural complexity in natural ecosystems, where heterogeneous habitats contribute to increased beta diversity across spatial scales. Our findings, supported by the estimation of ecological processes based on $\beta$NTI and $RC_{Bray}$ (Fig. 4b and Supplementary Fig. 10), illustrate that the role of homogeneous selection was stronger for community assembly in croplands, suggesting the consequence of agricultural conversion on homogeneous abiotic and biotic conditions across space. The impact of agriculture on biotic homogenization might vary at different scales. In contrast to our results, a regional survey on the conversion of steppe to cropland demonstrated that agriculture increased spatial heterogeneity of soil functional genes[29]. The lower functional turnover in steppe may be attributable to stable and similar soil environments across the region. Diverse in local but functionally homogeneous sward in regional natural steppe ecosystem exerts a stabilizing effect on the soil environment and soil ecosystem processes, reducing the impact of spatial and temporal variation in climate, soil texture and

topography[29]. Differently, agricultural management such as seasonal planting, crop types, and fallow cycles actually contribute to greater temporal and spatial variability that selects for greater heterogeneity across the region. Given the complexity of the soil environment, more attention needs to be paid to the biotic homogenization caused by agricultural conversion of the soil microbiome at various spatial scales.

Our results showed that land use change had a greater impact on taxonomic composition than on functional composition, highlighting the functional redundancy of soil microbiomes[35,63,64]. Soil microorganisms represent the most biologically and phylogenetically diverse community on Earth[65]. Although the taxonomic composition of soil microbiome varies tremendously across soil, microbial gene composition or functional capacity remains highly conserved[63,66], with lots of phylogenetically unrelated taxa carrying similar genes and performing similar functions[38]. For example, lignin substrate can be degraded by gram-negative bacteria *Comamonadaceae* and *Caulobacteraceae*, and the genus *Asticcacaulis* and *Caulobacter* (members of *Caulobacteraceae*) could degrade both hemicellulose and cellulose and all three lignocellulosic polymers, respectively[67]. Numerous microorganisms with the ability to participate in carbon degradation can coexist on the surface of plant residues[68]. Agricultural conversion, however, had minimal impact on overall carbon degradation and fixation, but did reduce nitrogen fixation and phosphorus mineralization and transportation potential (Fig. 3), suggesting the functional redundancy for carbon metabolism in soils. The fact that the potential for nitrogen fixation and phosphorus mineralization is reduced, indicates that croplands rely less on these processes due to the breakdown of nutrient cycling plant-microbial symbioses under agricultural fertilization. Taken together, our results indicated that agricultural land-use change significantly altered microbial taxonomic composition while the gene content remains relatively conserved, especially in relation to carbon metabolism. More realistic functional gene expression studies the functional divergences, redundancies, and complementarities in the different land use scenarios, e.g. metatranscriptomics[69] or quantitative stable-isotope probing (qSIP)[70], that correlate with the observed taxonomic shifts after agricultural conversion, needs to be further revealed in the future.

Changes in soil microbial communities across space are often strongly correlated with differences in soil abiotic and biotic conditions[47]. Similar to previous study[71], we observed soil pH is a major driver of the diversity and composition of soil bacterial communities across land-use types. More importantly, we found that fungal communities, particularly pathogens and saprotrophs, were strongly associated with changes in soil bacterial communities. Interactions between fungi and bacteria could partly drive the bacterial community shifts along a steep gradient of fungal community change[72,73]. For example, manipulating fungal richness can immediately mediate assembly processes of bacterial community[74]. The fungal hyphae could provide soil bacteria with ecological opportunities in severely carbon-limited soils by releasing carbonaceous compounds and providing a colonizable surface for the creation of new bacterial niches[72,73]. In addition to the effect of external conditions (e.g., biotic interactions and abiotic environmental conditions), our results also emphasize the important roles of microbial traits in regulating the response of microbial composition to agricultural conversion. The dormancy potential strategy changed from sporulation and toxin−antitoxin systems to resuscitation-promoting factors[42]. The sporulation trait affects species composition, with the abundance of phyla Firmicutes and Actinobacteria with spore-forming ability[75] increasing in croplands. The impact of regional species pools on cropland bacterial diversity is modulated by sporulation trait[55]. Many taxa with spore-forming ability had a higher species pool effect, indicating their survival and competitive advantage under environmental stress, as well as their retention during land use changes or their greater likelihood of spreading from natural ecosystems due to their adaptive capabilities.

Our findings provide a valuable insight for predicting ecological consequences of land-use change and agricultural management. The links between microbial composition and ecosystem function suggest that biotic homogenization have previously unrecognized and negative consequences for agricultural sustainability and service. Although the functional redundancy with C metabolism of soil microbiomes supports the stability and resilience of ecosystem functioning in response to perturbations[63], increased agricultural intensification gives rise to large uncertainty in predicting the loss of ecosystem function. It is also important to note the ways observations at different spatial scales can impact the interpretation of broad soil microbiome responses. Although our study covered a global scale, study sites and sequencing data were not evenly distributed. Most observations focus on forest-cropland ecosystem contrasts and are subject to methodological limitations arising from comparisons of sequencing methods and sampling schemes. Overall, our study suggests that biotic homogenization of the belowground microbiome across large spatial scale should be taken into account when evaluating the sustainability and soil health of agricultural management practices.

## Methods
### Continental survey and sampling
We conducted a continental field survey in croplands and adjacent natural ecosystems from 44 regions across China (Fig. 1a and Supplementary Table 1). Adjacent natural ecosystems were ~2 km from croplands and were selected to represent the most common and relatively undisturbed ecosystems, including forests, grasslands and wetlands. The distance between cropland and adjacent natural ecosystems is about 2 km in order to maintain a consistent climate and soil type. Among natural ecosystems of the 44 study regions, 30 regions include forests, grasslands, and wetlands, five regions include forests and wetlands, four regions include grasslands and wetlands, three regions include grasslands and wetlands, and two regions only include forests (Fig. 1c and Supplementary Table 1). The study survey represents a wide range of climate and soil gradients of climate, soil, and vegetation types (from tropical to boreal zones). For instance, mean annual precipitation and mean annual temperature in these regions are from 78 to 1775 mm and −2.8 to 24.4 °C, respectively. Soil pH ranged from 4.63 to 10.18 and soil organic matter ranged from 4.64 to 60.22 g·kg$^{-1}$ across all of the survey regions, representing broad environmental conditions.

To reduce variation between regions as much as possible, we focused on fields planted with maize (*Zea mays*) to represent agricultural systems since maize is widely cultivated throughout China and the world, with a total production exceeding that of wheat or rice[76]. In each region, we collected 4 to 10 plots of each ecosystem type. Composite surface soil samples (top ~20 cm depth) were collected at each plot in July and August 2019, during the crop growing season. Soil samples at each site pair were collected within one day to minimize the impact of sampling times. Each plot has a size of 2 × 2 m$^2$ and is the same across sites and ecosystems. We focused on surface soils because (1) topsoil is most affected by land use change; (2) agricultural management practices also primarily deal with topsoil, such as conventional tillage and crop root growth, which shape the tillage layer. In brief, soil samples were mixed by taking three soil cores with a 5-cm-diameter auger for each plot in the surface layer. After sampling, we thoroughly rinsed the soil auger using clean water. To ensure disinfection and sterilization, we then applied a 75% alcohol solution to its surface. Afterward, we placed the auger bit into a sterile bag for safekeeping until the subsequent sampling event. These soil samples were sieved through a 2.0-mm mesh to remove plant roots, litter, rocks, and other debris. A total of 1185 soil samples were collected representing 856 paired soils, with 303 forest-cropland pairs, 275 grassland-cropland pairs, and 278 wetland-cropland pairs obtained (Supplementary Table 2). Each soil sample was divided into two subsamples

where one set was frozen at −80 °C for DNA extraction and microbial analysis and the other set was air dried for measurement of soil physical and chemical properties.

## Global-scale meta-analysis

We conducted an extensive literature survey from 2013 to February 2023 using the Web of Science database (https://www.webofscience.com/). The format of the keywords used for the literature search includes (bacteri*) AND (land use change OR land cover change OR land use/cover change OR LULCC OR LUCC OR cropland OR farmland OR arable). After downloading the literature based on the keywords above, we obtained a total of 297 publications (Supplementary Fig. 12). Following the criteria below, we conducted the initial selection of the studies: (1) studies with a one-to-one correspondence of sequencing data between agricultural land and natural ecosystems were included; (2) articles for which sequencing metadata were not available from public repositories or upon request from individual study authors were excluded. After the initial selection, 75 studies were left, over 6000 sample sequencing data. In these studies, high-throughput sequencing of bacterial communities was conducted using Illumina, Ion S5, and 454 pyrosequencing platforms. Twenty-three primer pairs were identified from the research metadata, and the most used primers in the sample were 515F and 907R (18/75), 515F and 806R (13/75), and 338F and 806R (16/75). After downloading the raw data corresponding to the data availability provided in the articles, the raw sequences were processed using QIIME 2 and annotated using the USEARCH tool. A final ASV dataset comprising 3482 samples was remained for subsequent analysis after excluding low-reads (<10,000 reads) and low-quality samples. We utilized the -fastq_filter command in the vsearch tool for sequence quality control, with the parameter -fastq_maxee set to 1. This implies that the maximum expected errors threshold for low-quality bases in all sequences is set to 1. Only sequences with an expected error count less than or equal to 1 are retained, while sequences exceeding this threshold are filtered out. In addition to sequencing data, we also collected the following parameters: ecosystem type, plant type, location (i.e., latitude and longitude). Taking into account sample size and coverage, we selected forest to represent natural ecosystem because forest soils included more than 1300 samples and covered six continents. Other ecosystems with only a few sites or low distribution range lacked representation for large-scale evidence (Supplementary Fig. 13), and were excluded from further analyses. In total, 2403 samples were included in the global-scale meta-analysis.

## Soil environmental variables

We evaluated soil chemistry and nutrients to gauge changes across agricultural land-use change and to consider the implications of those variables on microbial communities. Here, we selected the most important six soil variables, i.e., soil pH, organic matter (OM), soil moisture (Mo), available phosphorus (AP), and available nitrogen ($NO_3$–N and $NH_4$–N). These indicators were recognized as the main soil variables influencing bacterial diversity patterns at global and regional scales[77–79]. Soil pH was assessed in a 1:5 suspension (soil to distilled water) using a pH meter. Organic matter was determined calorimetrically following oxidation with a combination of potassium dichromate and sulfuric acid. Soil moisture was measured by the gravimetric method after samples were oven-dried at 100 °C for 24 h. $NO_3$-N and $NH_4$-N concentrations were measured using 1 M KCl solution with Continuous-Flow AutoAnalyzer. Available phosphorus concentrations were extracted by $NaHCO_3$ and measured by molybdenum blue colorimetry. We measured soil physicochemical properties for each plot. Local soil filtering was calculated as the average of all plots within each ecosystem for each soil variable and local soil heterogeneity was calculated as the within-ecosystem standard deviation of each soil variable.

## Soil enzyme activities

The activities of soil extracellular enzymes involved in C, N, and P acquisition were determined using the microplate-scale fluorometric method[80]. We used a 200 μM solution of substrates labeled with 4-methylumbelliferone or 7-amino-4-methylcoumarin. The C-acquisition enzymes analyzed included $\beta$–1,4-glucosidase (BG), 1,4-$\beta$-Dcellobiohydrolase (CBH) and $\beta$-xylosidase (BX). The N-acquisition enzymes analyzed were $\beta$–1,4-N-acetylglucosaminidase (NAG) and L-leucine aminopeptidase (LAP), while the P-acquisition enzyme analyzed was alkaline phosphatase (APP). After incubation at 35 °C, plates were centrifuged, and the supernatant was transferred to black, flat-bottom 96-well plates. Fluorescence was measured using a microplate reader with 365 nm excitation and 450 nm emission filters. Soil enzyme activities were expressed as nmol g$^{-1}$ dry soil h$^{-1}$.

## DNA extraction, amplicon sequencing, and data preprocessing

Genomic DNA was extracted from 0.5 g of the soils using the MP FastDNA spin kit for soil (MP Biomedicals, Solon, OH, USA) according to the manufacturer's instructions. The diversity of soil bacteria and fungi was measured by 16 S rRNA gene and nuclear ribosomal ITS amplicon sequencing using an Illumina MiSeq PE250 platform. For the bacterial community, 16 S rRNA genes were amplified using primer set 515 F (5′-GTGCCAGCMGCCGCGGTAA-3′) and 907 R (5′-CCGTCAATTCCTTTG AGTTT-3′), targeting the V4-V5 region of the 16 S rRNA gene. For the fungal community, the first nuclear ribosomal ITS sequences were amplified using primers ITS5-1737F (5′-GGAAGTAAAAGTCGTAACAAGG-3′) and ITS2-2043R (5′-GCTGCGTTCTTCATCGATGC-3′), targeting the ITS1-5F region. PCR amplification was performed in a 50 μl volume: 25 μl 2x Premix Taq (Takara Biotechnology, Dalian Co. Ltd., China), 1 μl each primer (10 μM) and 3 μl DNA (20 ng/μl) template. The PCR thermal cycling conditions were performed by thermocycling: 5 min at 94 °C for initialization, followed by 30 cycles of 30 s denaturation at 94 °C, 30 s annealing at 52 °C, 30 s extension at 72 °C, and 10 min final elongation at 72 °C. The length and concentration of the PCR product were detected by 1% agarose gel electrophoresis. Sequencing libraries were generated using NEBNext® Ultra™ II DNA Library Prep Kit for Illumina® (New England Biolabs, MA, USA) following the manufacturer's recommendations and index codes were added. Bioinformatic processing, including filtering, dereplication, sample inference, chimera identification, and merging of paired-end reads, was performed using the Divisive Amplicon Denoising Algorithm 2 (DADA2) package in R[81]. In brief, the plotQualityProfile command was run to detect the quality of the amplified sequences. We imposed a minimum length of 100 bp to remove any small fragments at the filtering stage, at which, the error in the maxEE argument was 2 as this optimized the retention of reads throughout the pipeline. Error rates were subsequently calculated by the DADA2 algorithm before dereplication and merging of paired end sequences. Chimeras were removed using the removeBimeraDenovo command with method = "consensus"[82]. Finally, the taxonomical annotation of the representative sequences of amplicon sequence variants (ASVs) was performed with a naïve Bayesian classifier using the Silva v. 138 (for bacteria) and the UNITE v. 7 (for fungi) database[83,84]. It should be noted that although the ITS region is by far the best option as a general DNA (meta) barcoding marker for fungi, there are inherent limitations associated with the use of a ITS region for enabling in-depth characterization of fungal communities. We were not concerned with changes at the fungal species level, so ITS region sequencing should have limited impact on our results. The sequence number in each sample was rarefied to the same depth for the 16 S rRNA gene (15000 reads) or ITS sequences (21921 reads), leaving a total of 31,402 bacterial ASVs and 77,962 fungal ASVs for further analyses.

## Shotgun metagenome sequencing

A subset of 40 samples from 10 regions covering cropland, forest, grassland and wetland soils were selected for metagenomic

sequencing to analyze changes in microbial community functional potential ($n = 10$ per ecosystem type; Supplementary Fig. 1a). Metagenomic libraries for 40 samples were prepared according to the product instructions of ALFA-SEQ DNA Library Prep Kit (Findrop, Guangzhou, China) and index code was added. Initial quantification of the library concentration was performed using Qubit 3.0 fluorometer (Life Technologies, Carlsbad, CA, USA) and the library was diluted to 1 ng/μL. Agilent 2100 Bioanalyzer System (Agilent Technologies, CA, USA) was used to detect the integrity of library fragments and the length of insert size. Then, the library was sequenced on Illumina Novaseq 6000 platform (Illumina, San Diego, CA, USA) to generate 150 bp paired-end reads at Guangdong Magigene Biotechnology Co., Ltd. In total, $1.71 \times 10^9$ raw reads were sequenced across all samples, which yielded 512.3 Gbp of total sequence information with an average data volume of 12.8 Gbp per sample. Raw data were quality checked with FastQC (v0.11.9) and processed using Trimmomatic v.0.39 (leading: 3, trailing: 3, slidingwindow: 4:15, minlen:36) to trim adapters and discard bases with a quality score <15 and length <36 bp. After that, 12.2 Gbp clean data per sample were obtained. Clean reads were annotated for functional analysis of the microbiome using HUMAnN v3.7 (based on DIAMOND (version 2.1.6)[85] and Bowtie2 (version 2.5.1)[86]) with ChocoPhlAn database (version "mpa_vJan21_CHOCOPhlAnSGB_202103") and UniRef90 (version "uniref90_201901b") protein database to quantify relative abundance of functional genes and metabolic pathways[87]. The annotation results were organized according to Kyoto Encyclopedia of Genes and Genomes (KEGG) Orthologues (KOs), Clusters of Orthologous Group of proteins (COG) functional categories and MetaCyc functional pathways using "humann3_regroup_table" script. The abundance of functional gene was expressed as Transcripts per million.

## Estimation of ecological processes

The estimation of ecological processes was performed according to Stegen et al.[88]. The aim of framework is to quantitatively estimate the degree to which spatial turnover in community composition is influenced by selection, drift acting alone, dispersal limitation acting in concert with drift and homogenizing dispersal. The estimation of ecological processes followed a two-step procedure. First, we quantified $\beta$NTI ($\beta$-nearest taxon index) for all pairwise community comparisons. A value of $|\beta$NTI$| > 2$ indicates that observed turnover between a pair of communities is governed primarily by selection. A value of $|\beta$NTI$| < 2$ indicates that observed turnover between a pair of communities is governed by drift, dispersal limitation and homogenizing dispersal. $\beta$NTI $< -2$ indicates significantly less phylogenetic turnover than expected (i.e., homogeneous selection) while $\beta$NTI $> 2$ indicates significantly more phylogenetic turnover than expected (i.e., variable selection). Second, we quantified Raup–Crick (RC$_{bray}$) for pairwise community comparisons that were not governed by selection (that is, those with $|\beta$NTI$| < 2$). The relative influence of homogenizing dispersal was quantified as the fraction of pairwise comparisons with $|\beta$NT$| < 2$ and RC$_{Bray} < -0.95$. Dispersal limitation was quantified as the fraction of pairwise comparisons with $|\beta$NTI$| < 2$ and RC$_{Bray} > 0.95$. The fractions of all pairwise comparisons with $|\beta$NTI$| < 2$ and $|$RC$_{Bray}| < 0.95$ were used to estimate influence of "undominated" assembly, which mostly consists of weak selection, weak dispersal, diversification, and/ or drift[89]. $\beta$NTI and RC$_{Bray}$ could differentiate the relative importance of five assembly processes to the whole community. The five assembly processes were assessed for their relative importance in governing community variations under agricultural land-use change.

## Statistical analyses

All statistical analyses were conducted in the statistical platform R (V4.2.1; http://www.r-project.org/; Supplementary Table 4).

Large-scale microbial homogenization was reflected by a decrease in community turnover rate (decreased $\beta$-diversity in space). To analyse the response of $\beta$-diversity to agricultural conversion, we calculated taxonomic (16S) and functional (KEGG and COG module level) community dissimilarity between sites using Bray–Curtis index. We tested the effects of agricultural impacts on the relative abundance of microbial taxonomic and functional groups using linear mixed-effects model (LMM), in which sites were termed as random intercept effects. Microbial functional groups were predicted by the Functional Annotation of Prokaryotic Taxa (FAPROTAX)[35] and PICRUSt2[90]. Analysis of LMM was conducted in lme4 R packages[91]. To characterize how microbial communities differ, Principal coordinate analyses (PCoA) were conducted on Bray–Curtis index to examine dissimilarities among taxonomic and functional composition between croplands and natural ecosystems. PERMANOVA was utilized to test the statistical significance of dissimilarity among ecosystem types. To link soil environmental and fungal variables to microbial communities, the correlations between soil filtering and heterogeneity and fungal functional groups were tested by Mantel correlations. Fungal phylotypes were assigned into three functional groups—soil saprotrophs, litter saprotrophs and plant pathogens using FungalTraits[92]. To assess changes in functional genes with agricultural conversion, we calculated log2-fold changes in croplands relative to natural ecosystems (forests, grasslands, and wetlands) using DESeq2 with the apeglm shrinkage algorithm. We also used DESeq2 to identify microbial phylotypes, and functional gene annotation assigned to COG and KEGG that significantly increased, decreased and unchanged under agricultural impacts relative to natural ecosystems.

To discern the direct and indirect effects of agricultural impacts on microbial composition and soil functions, a structural equation model was conducted to assess the causal relationships among agricultural land-use change, soil environmental variables, fungal communities, and microbial composition and soil functions. We first considered a hypothesized conceptual model (Supplementary Fig. 11) that included all reasonable pathways. Then, we sequentially eliminated non-significant pathways unless the pathways were biologically informative or added pathways on the basis of the residual correlations[75]. Three metrics were used to quantify the goodness of fit of SEM models: the $\chi^2$ test, the root mean square error of approximation (RMSEA), and the Comparative Fit Index (CFI). Specifically, the closer to 1 CFI value, closer to 0 RMSEA values, and the higher $\chi^2$ and RMSEA $P$ values, the better model performs. With a good model fit, we were able to interpret the path coefficients of the model and their associated $P$ values. A path coefficient is analogous to the partial correlation coefficient, and describes the strength and sign of the relationship between two variables. Microbial taxonomic composition (16S) and functional (KEGG) composition were represented by the principal coordinate analyses 1, the first component of PCoA analysis. SEM were conducted using 40 site samples in the "lavaan" package in R environment[93].

## Reporting summary

Further information on research design is available in the Nature Portfolio Reporting Summary linked to this article.

## Data availability

All data required to reproduce the results are available in the Figshare Database (https://doi.org/10.6084/m9.figshare.25396525). The raw sequence data that support the findings of this study are openly available in the Beijing Institute of Genomics (BIG) Data Center, Chinese Academy of Sciences, under BioProject accession no. PRJCA020242 (16S amplicon) and PRJCA020245 (Metagenomics) and are publicly accessible at http://bigd.big.ac.cn/gsa. Source data are provided with this paper.

## Code availability

All scripts are available on GitHub (https://github.com/Pong2021/Agricultural-impacts-on-soil-microbiome-function.git) and Zenodo[94].

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

## Acknowledgements

This work was supported by the National Science Foundation for Excellent Young Scholars of China (grant No.: 42122050; S.J.), the National Key Research and Development Program of China (grant No.: 2021YFD1900500; S.J.), Joint Fund of the National Natural Science Foundation of China (grant No.: U21A2029; G.W.), and National Science Foundation of China (grant No.: 42077222; S.J.).

## Author contributions

All authors contributed intellectual input and assistance to this study and the manuscript preparation. Z.P. conducted the experiments, analyzed the data, and wrote the manuscript. G.W. and S.J. conceived and designed the experiments, and revised the manuscript. M.G.A.v.d.H. helped with data analysis, interpretation and revision of the manuscript. Y.L., H.G. and S.C. contributed to field experiment. X.L., Y.A., J.Q., H.P., B.C. and C.L. contributed to survey collection and sample processing. X.Q., L.J. and Y.Z. contributed to metagenomic processing.

## Competing interests

The authors declare no competing interests.
