## [Peer Review File · Nature Communications]

Reviewers' Comments:

Reviewer #1:

Remarks to the Author:

Peng et al. investigated the impacts of agricultural land-use changes on the taxonomic and functional biotic homogenization through a survey of soil microbial communities in over 1,100 samples. We all know that land-use change could cause biotic homogenization, but this notion has not been confirmed in belowground communities. By a large-scale survey using high-throughput sequencing and metagenomic techniques combined with a global scale meta-analysis, they confirmed the agriculture-induced homogenization of belowground composition and functions. Next, they focused on shifts in specific taxonomic groups and functions that caused homogenization and revealed how agriculture alters soil functions via changes in microbial communities. Finally, the authors unraveled the comprehensive impact of agriculture through field experiments. By setting soil management, exogenous inputs and crop management, the author explored the specific process of the comprehensive agricultural impact on soil functions. A distinctive feature of this manuscript is the sampling and analysis of continental-scale soil microbiota data as well as global scale meta-analysis and field experiment to study agriculture-induced belowground biotic homogenization and its associated microbial compositional and functional changes. This paper analyses an outstanding dataset and multiple approaches, and can be a good reference of how agriculture alters belowground communities and functions. I believe that this article has a strong potential to be impactful. Despite that, I am concerned about the condensation and generalization of some results in this paper, which does not convey the work well. In my opinion, as soon as those aspects are enhanced, this paper will attract a wide readership.

One concern is the abstract does not well summarize and organize the major findings in the main text. First, why set up a field experiment to simulate different agricultural management? This requires giving specific experimental treatment and research aim to help better understand the experiment in the abstract. This can also provide further supports for summarizing the results of field experiments. I guess the author wants to use field experiments to separate the comprehensive impact of agriculture. For example, are the shifts in soil respiration caused by agriculture primarily contributed by soil management or fertilization? Second, an important result is that which major taxa are affected by agriculture is not shown in the abstract. Large-scale surveys could compensate for randomness and bias at individual sampling sites and provide general patterns of agricultural impacts. Finally, the authors found an interesting phenomenon, in which taxonomic composition caused by land use was largely related to fungal communities rather than soil pH, which is associated with functional composition in study. It is well known that soil pH is a major driver of biogeographical distribution of bacterial phylogeny. Soil pH was not the main driver under land use changes, probably because changes in soil pH were small and fungal communities were closely related to agriculture-induced plant communities. This interesting phenomenon need a deeper explanation in the main text and can be presented in the abstract. In addition, the statistical methods section needs to be moderately expanded and/or clarified. There are some tests performed throughout the paper that are not explained in the methods and some methods need to be expanded. Once these issues are addressed, I believe that this article could be a great contribution to revealing the consequences of land-use change on soil microbiome function.

Minor comments:

L68-70: Poor sentence, please re-write this sentence.

L72-74: Too long sentence, split into two sentences.

L75: The aim of field experiment needs to be supplemented to better understand the link between large-scale sampling and field experiment.

L 84: "These results would advance..."

L231-236: The impact of fungal communities and soil pH on taxonomic composition and functional composition, respectively, requires more discussion.

L337: Soil parameter ranges are interesting. Please give it was from mean values of 44 regions or from all >1100 plots? With plots, the range is perhaps not so informative if there are outliers.

L339: How large can be soil type variation within a region (e.g. due to topography, base rock etc.).

L359: Please provide the specific coordinates of the field experiment conducted.

L369: Please give reasons for conducting these three treatments. Are these three management

practices representative of agricultural activities?

L371: Are forests and meadows at the same altitude?

L 425: What is the data volume of each metagenome?

L 438: Need more details for the methods of ecological processes estimation.

L449: Since all the statistical analysis was performed in R, it would be nice to see a table in supplemental with the list of packages used for each analysis to help increase the transparency of their analytical approach.

Figure 2: X-axis lacks effect size in linear mixed-effects models.

Figure S1: Units of latitude and longitude are missing.

Reviewer #2:

Remarks to the Author:

The authors set out to examine how agricultural land use change from various natural habitats impact soil microbial community structure and putative function with: (1) a comprehensive global-scale meta-analysis, and (b) a China-wide field survey of crops and adjacent natural ecosystems with a paired field experiment of agricultural conversion.

The methods are largely robust, with some minor feedback, but my main concern is the lack of separation or cross-synthesis of the global meta-analysis and China-only study. Maybe this should be separated into two studies, or please make it clearer how the studies *cohesively* fit into one study.

The major concern with this study is the direct use of functional annotation of metagenomic reads to infer changes in function. Other than bulk biogeochemistry (OM, available phosphorus, nitrogen), there are no methodological details of how the enzyme assays were conducted, so I am assuming these were inferred from the metagenomic data (KEGG, and KO annotation)? If so, this is an assumption that should be tested before inferring functional changes in the microbial community (See Rocca et al 2015 ISME, Bier et al 2015 FEMS, Hall et al 2018 Nat Microbiol). Further, linking changes in microbial genera to the functional shifts is a bit misleading, as the data are both originated from DNA, hence a correlation would be expected. Actual enzyme assays would be more robust correlations with the microbial community data, if they exist.

Line-by-line feedback:

59 - 'Soil microbiomeS...'

326 - Please organize the methods as the rest of the paper is presented - continental-scale analysis, then the paired crop/natural habitat within China. Or, flip the rest of the paper to match the methods order.

339 - how was this confirmed? Please present data to indicate how similar the conditions are for the crop and adjacent natural habitat. Climate may likely be similar on ~2km scale, but water availability, etc, could easily vary across that spatial scale.

345 - I assume the site pairs were collected within short timescales? Please confirm this timeframe.

394 - Please define 'available' - do you mean bio-available?

426-437 (261-266) - Other than bulk biogeochemistry (OM, available phosphorus, nitrogen), there are no methodological details of how the enzyme assays were conducted, so I am assuming these were inferred from the metagenomic data (KEGG, and KO annotation)? If so, this is an assumption that should be tested before inferring functional changes in the microbial community (See Rocca et al 2015 ISME, Bier et al 2015 FEMS, Hall et al 2018 Nat Microbiol). Further, linking changes in microbial genera to the functional shifts is a bit misleading, as the data are both originated from

DNA, hence a correlation would be expected. Actual enzyme assays would be more robust correlations with the microbial community data, if they exist.

437 - why is this abbreviation made if it is only mentioned once?

715 - Don't you mean 'dissimilarities between cropland and *natural ecosystems*.'?

Figs:

Please list the actual figures *before* the extended data figs. I realize these would be properly organized in any publication, but it is confusing for the reviewers, as I saw the smaller subset of China samples before seeing the actual Figure 1.

Across all meta-analysis effects, please graphically depict what is being compared (e.g. Fig 1, d, h; Fig 2 a, c; Extended Fig 7). For example, in Fig 2b/d, upper right indicates a which conversion/comparison is being made.

Fig 1:

Panel c - Define 'All natural ecosystems' - meaning no cropland-conversion sites, or given the photo expansion, all four habitat categories are present at this location? Unclear what the panel C photos indicate? The cropland bracket includes all categories, but for 'All natural ecosystems' does this mean there are/are not close habitat pairings of cropland?

Panel d, h: Please indicate the sample number for each meta-analysis effect.

Extended Data Fig 1 - please further clarify this subset of points, vs. the full dataset presented in Fig 1.

Extended Fig 6 - This figure is not useful, aside from conveying that microbial genera have overlapping functional pathways. Please consider another graphical display if this is to be included.

Reviewer #3:

Remarks to the Author:

This study by Peng and colleagues aimed to elucidate taxonomic and functional homogenization due to intensive agriculture. The authors used a combination of meta-analysis, large-scale field study, and field trial experiments to assess the composition of soil bacterial and fungal communities. They found that agricultural land-use had a strong effect on microbial composition. While biotic homogenization is certainly an important area of research that deserves more attention, I do not think this study clearly shows that. The three studies used here are quite different from each other in terms of experimental design and questions. While they show compositional differences mediated by landuse, there is no coherent evidence that suggests homogenization. The manuscript also suffers from some critical flaws and inconsistencies. Please see my detailed comments below.

Introduction: the introduction is quite weak. The first two paragraphs are full of broad sentences and do not lay the foundation of this study. If biotic homogenization is the focus, I feel the introduction can be more nuanced, e.g., what are the signatures and drivers of homogenization at local, regional, and global scales? how can we identify homogenization? is it only the loss of richness? I think this would help readers understand the importance of this study. Question 2 is not novel and has been assessed by many previous studies. Were there any specific hypotheses you tested?

Homogenization: I am confused regarding the homogenization indicator/metric used in this study. I am not sure if readers would agree with the main claim. Except for Figure 1b panel, the rest of the results show taxonomic and functional differences between land-use types. Sure, that information is also important, but this has been demonstrated by many studies. I was also looking

for clear evidence of homogenization.

Field experimental design: I could not understand the rationale and design of this experiment. What were the rationale and hypotheses behind imposing tillage, monoculture and fertilizer applications to forest and grassland soils? How could these results be compared with arable soils? What was the control in this experiment and how was it different from initial forests/grasslands? Were these plots in the same location? The authors say 27 in forest soils and 27 in grassland soils. What was the tillage type? Overall, this is a confusing experiment with 9 different agricultural treatments imposed on forest and grassland soils.

Methods

Ln 344: A major limitation of this study is the number of plots, which varied considerably between ecosystem types. When samples are combined, this could impact the level of microbial diversity observed for each site. This could also have a confounding effect on the results. I wonder why same number of plots were not sampled for all ecosystem types. Further, the plot size is also missing. Did the plot size also vary between sites and ecosystems?

Number of soil samples: how did you come up with 1185 samples (Ln 354)? Abstract says 856 paired soils. The supplementary Information section needs more development. Figure captions are inadequate.

Fungal ASVs: The authors amplified the ITS region to assess fungal ASVs. Due to the inherent variability of ITS region, as shown in many recent studies, I do not think ITS data can be used to obtain amplicon sequence variants (ASVs). Even variation of a single base might produce a new ASV. I am not sure if fungal ASV data are admissible. There are also some inconsistencies and missing information. What was the total number of ASVs for 16S and ITS?

Amplicon and metagenomic sequencing: This section is missing a lot of relevant details. The authors should add information related to various steps (e.g., library preparation, trimming, merging,). What kind of MiSeq sequencing did you perform (250, 300 paired end)? What were the controls? For metagenomics, how were the 10 samples chosen for each ecosystem type?

Statistical analyses: The analysis performed in this study is unbalanced and under-described. For functional annotation, they used fungal ITS. However, for statistical analyses (e.g., beta-diversity, SEM), they only used bacterial 16S. This is odd! Also, please add the number of samples for important analyses. For example, what was the total number of samples for SEM?

Meta-analysis: Ln 507- what proportion of samples were sequenced using 515F and 806R? Ln 511- what was the quality threshold? So, out of 3482 samples, 1300 were forest soil samples? I see that other ecosystems were removed from further analyses. So, it is important they call it forest and not 'natural ecosystems'.

Methods shows they only used FungalTraits, but then I also see FAPROTAX. Which samples were annotated with FAPROTAX?

Results

Figures: Finding the main figures was quite difficult. There is a whole extended data section between figure legends and the main figures, so it was challenging to understand the figures and source of samples. As there are three studies in this study, figures should be clearly marked, i.e., results from which study they are showing specifically. It is important that authors describe figures categorically and not combine everything, as the three studies are very different in terms of design and settings.

Discussion

There is hardly any discussion! It is similar to the abstract in terms of length. For a manuscript with three large studies, this section is insufficient and should be rewritten.

Response and actions taken with respect to the reviewer comments for:

NCOMMS-23-43840-T

Title: Large-scale evidence of agriculture-induced taxonomic and functional
homogenization of belowground communities

Authors: Ziheng Peng ¹, Xun Qian ², Yu Liu ¹, Xiaomeng Li ¹, Hang Gao ¹, Yining An
¹, Jiejun Qi ¹, Lan Jiang ², Yiran Zhang ², Shi Chen ¹, Haibo Pan ¹, Beibei Chen ¹,
Chunling Liang ¹, Marcel G.A. van der Heijden ^{3,4}, Gehong Wei ^{1*}, Shuo Jiao ^{1*}

Reviewer #1 (Remarks to the Author):

Peng et al. investigated the impacts of agricultural land-use changes on the taxonomic and functional biotic homogenization through a survey of soil microbial communities in over 1,100 samples. We all know that land-use change could cause biotic homogenization, but this notion has not been confirmed in belowground communities. By a large-scale survey using high-throughput sequencing and metagenomic techniques combined with a global scale meta-analysis, they confirmed the agriculture-induced homogenization of belowground composition and functions. Next, they focused on shifts in specific taxonomic groups and functions that caused homogenization and revealed how agriculture alters soil functions via changes in microbial communities. Finally, the authors unraveled the comprehensive impact of agriculture through field experiments. By setting soil management, exogenous inputs and crop management, the author explored the specific process of the comprehensive agricultural impact on soil functions.

A distinctive feature of this manuscript is the sampling and analysis of continental-scale soil microbiota data as well as global scale meta-analysis and field experiment to study agriculture-induced belowground biotic homogenization and its associated microbial compositional and functional changes. This paper analyses an outstanding dataset and

multiple approaches, and can be a good reference of how agriculture alters belowground communities and functions. I believe that this article has a strong potential to be impactful. Despite that, I am concerned about the condensation and generalization of some results in this paper, which does not convey the work well. In my opinion, as soon as those aspects are enhanced, this paper will attract a wide readership.

Response: Thank you very much for the positive comments. We have carefully revised the manuscript and condensed our results for clarity.

One concern is the abstract does not well summarize and organize the major findings in the main text. First, why set up a field experiment to simulate different agricultural management? This requires giving specific experimental treatment and research aim to help better understand the experiment in the abstract. This can also provide further supports for summarizing the results of field experiments. I guess the author wants to use field experiments to separate the comprehensive impact of agriculture. For example, are the shifts in soil respiration caused by agriculture primarily contributed by soil management or fertilization?

Response: The manuscript contains indeed a lot of data already. However, to focus on the main results of the meta-analysis and the continental scale analysis, we have now removed this part from the manuscript. If the editor or other reviewers feel we should keep this part in, we are happy to add it back again.

Second, an important result is that which major taxa are affected by agriculture is not shown in the abstract. Large-scale surveys could compensate for randomness and bias at individual sampling sites and provide general patterns of agricultural impacts.

Response: Thanks for your suggestion. We have now added the most important taxa that are affected by changes in land use in the Abstract section, “Specifically, croplands were enriched in Chloroflexi, Gemmatimonadota, Planctomycetota, Myxococcota and Latescibacterota.” (see L.35-36).

Finally, the authors found an interesting phenomenon, in which taxonomic composition

caused by land use was largely related to fungal communities rather than soil pH, which is associated with functional composition in study. It is well known that soil pH is a major driver of biogeographical distribution of bacterial phylogeny. Soil pH was not the main driver under land use changes, probably because changes in soil pH were small and fungal communities were closely related to agriculture-induced plant communities. This interesting phenomenon need a deeper explanation in the main text and can be presented in the abstract.

Response: We agree that this is an important result. However, due to the word limit, it is not possible to include this in the abstract and we further highlighted this in the main text (see L.357-368).

In addition, the statistical methods section needs to be moderately expanded and/or clarified. There are some tests performed throughout the paper that are not explained in the methods and some methods need to be expanded. Once these issues are addressed, I believe that this article could be a great contribution to revealing the consequences of land-use change on soil microbiome function.

Response: In the Methods, we added detailed information in experimental design and statistical analysis. We have added more detailed information of ‘Soil enzyme activities’, ‘DNA extraction, amplicon sequencing and data preprocessing’, ‘Shotgun metagenome sequencing’, ‘Estimation of ecological processes’ and ‘Statistical analyses’. These methods are stated in detail, and we have added instructions in the figure legends to improve the readability of the figures.

Minor comments:

L68-70: Poor sentence, please re-write this sentence.

Response: We have rewritten the Introduction and added a large section of introductions related to land-use change. We adjust this sentence to “Currently, we still lack a generalizable and consistent understanding of how soil microbial taxonomic and functional profiles respond to agricultural intensification and which microbial lineages and functions are mostly impacted across a wide range of soil and climate types ¹. This

knowledge gap hinders our comprehensive understanding of the global decline in biodiversity” (L.82-86).

L72-74: Too long sentence, split into two sentences.

Response: We have separated dataset of global-scale and continental scale to reveal the question for clarity, “We combined a global-scale meta-analysis and a continental soil survey to address these questions ... To gain a global perspective on agricultural-induced biotic homogenization, and to complement the continental scale soil survey, we also collected 16S rRNA amplicon-based sequencing data from soil samples of global agricultural-natural ecosystem pairs from all available gene banks (Fig. 1a)” (L.91-100).

L75: The aim of field experiment needs to be supplemented to better understand the link between large-scale sampling and field experiment.

Response: As mentioned above, we have removed the section of field experiment.

L 84: “These results would advance...”

Response: Yes, revised as suggested.

L231-236: The impact of fungal communities and soil pH on taxonomic composition and functional composition, respectively, requires more discussion.

Response: We have added a deeper explanation and discussion in the Discussion, “More importantly, we found that fungal communities, particularly the pathogen and saprotroph, are strongly correlated with changes in soil bacterial communities ... The fungal hyphae could provide soil bacteria with ecological opportunities in severely carbon-limited soils by releasing carbonaceous compounds and providing a colonizable surface for the creation of new bacterial niches ^{2, 3}” (L.360-368).

L337: Soil parameter ranges are interesting. Please give it was from mean values of 44 regions or from all >1100 plots? With plots, the range is perhaps not so informative if

there are outliers.

Response: Soil parameter was from all 1085 plots. We replaced values from plots with mean values of 44 regions, “Soil pH ranged from 4.63 to 10.18 and soil C ranged from 4.64 to 60.22 g·kg⁻¹ across all of the survey regions, representing broad environmental conditions”. (L.406-408)

L339: How large can be soil type variation within a region (e.g. due to topography, base rock etc.).

Response: There should be little difference in these factors given that the samples of each region were collected within 2 km, as shown in L.397-399.

L359: Please provide the specific coordinates of the field experiment conducted.

Response: The agricultural management practices field experiment site was established at the northern of the Qinling Mountains in June 2022 and located at the coordinates 34°13' N and 108° 04' E. However, to focus on the main results of the meta-analysis and the continental scale analysis, we have now removed this part from the manuscript. If the editor or other reviewers feel we should keep this part in, we are happy to add it back again.

L369: Please give reasons for conducting these three treatments. Are these three management practices representative of agricultural activities?

Response: As mentioned above, we have removed the section of field experiment. If the editor or other reviewers feel we should keep this part in, we are happy to add it back again. We acknowledge that there are many agricultural management practices, and also different between industrial and conservation agriculture. It is not possible to include all agricultural management practices, but these practices almost fall under crop management (i.e., crop rotations, cover cropping, residue retention, intercropping, agroforestry), external input (i.e., mineral and organic fertilizers, organic amendments, pesticides, biologicals) and soil management (i.e., shallow and deep tillage practices). Therefore, we select three common agricultural practices to be representative of

agricultural activities.

L371: Are forests and meadows at the same altitude?

Response: Yes, forests and grasslands are at the same altitude in field experiment.

L 425: What is the data volume of each metagenome?

Response: In total, 1.71×10^9 raw reads were sequenced across all samples, which yielded 512.3 Gbp of total sequence information with an average data volume of 12.8 Gbp per sample. We have added this information in the Methods (L.538-540).

L 438: Need more details for the methods of ecological processes estimation.

Response: We have added more detailed information for the methods of ecological processes estimation, “The estimation of ecological processes was performed according to Stegen et al. ⁴. The aim of framework is to quantitatively estimate the degree to which spatial turnover in community composition is influenced by selection, drift acting alone, dispersal limitation acting in concert with drift and homogenizing dispersal. The estimation of ecological processes follow a two-step procedure. First, we quantified β NTI (β -nearest taxon index) for all pairwise community comparisons. A value of $|\beta$ NTI|>2 indicates that observed turnover between a pair of communities is governed primarily by selection. A value of $|\beta$ NTI|<2 indicates that observed turnover between a pair of communities is governed by drift, dispersal limitation and homogenizing dispersal. β NTI < -2 indicates significantly less phylogenetic turnover than expected (i.e., homogeneous selection) while β NTI > 2 indicates significantly more phylogenetic turnover than expected (i.e., variable selection). Second, we quantified RC_{bray} for pairwise community comparisons that were not governed by selection (that is, those with $|\beta$ NTI|<2). The relative influence of homogenizing dispersal was quantified as the fraction of pairwise comparisons with $|\beta$ NTI| < 2 and $RC_{\text{Bray}} < -0.95$. Dispersal limitation was quantified as the fraction of pairwise comparisons with $|\beta$ NTI| < 2 and $RC_{\text{Bray}} > 0.95$. The fractions of all pairwise comparisons with $|\beta$ NTI| < 2 and $|RC_{\text{Bray}}| < 0.95$ were used to estimate influence of “undominated” assembly, which mostly

consists of weak selection, weak dispersal, diversification, and/or drift⁵. β NTI and RC_{Bray} could differentiate the relative importance of five assembly processes to the whole community. The five assembly processes were assessed for their relative importance in governing community variations under agricultural land-use change” (L.553-575).

L449: Since all the statistical analysis was performed in R, it would be nice to see a table in supplemental with the list of packages used for each analysis to help increase the transparency of their analytical approach.

Response: Thank you for kind comments. We have added a table list in the supplemental material to show the analysis methods and corresponding R packages.

Supplementary Table 4. The list of software and algorithms used for each analysis.

Analysis methods	Software and algorithms	Source
Estimating of ecological processes	https://github.com/Pong2021/Agricultural-impacts-on-soil-microbiome-function.git	
Calculation of beta diversity	R package vegan	6
Linear mixed-effects model	R package lme4	7
DESeq2	R package DESeq2	8
SEM models	R package lavaan	9
FAPROTAX	FAPROTAX database	10
FungalTrait	FungalTrait database	11

Figure 2: X-axis lacks effect size in linear mixed-effects models.

Response: We have added the label ‘effect size’ in Fig. 2.

Figure S1: Units of latitude and longitude are missing.

Response: We have added units of latitude and longitude in Fig. S1.

Reviewer #2 (Remarks to the Author):

The authors set out to examine how agricultural land use change from various natural habitats impact soil microbial community structure and putative function with: (1) a comprehensive global-scale meta-analysis, and (b) a China-wide field survey of crops and adjacent natural ecosystems with a paired field experiment of agricultural conversion.

The methods are largely robust, with some minor feedback, but my main concern is the lack of separation or cross-synthesis of the global meta-analysis and China-only study. Maybe this should be separated into two studies, or please make it clearer how the studies *cohesively* fit into one study.

Response: Thank you very much for the positive comments. To examine whether and how land-use intensification affects soil microbiome composition and functions, we combined three datasets and experiments at different scales (a global-scale meta-analysis and a continental field survey in China). Two experiments were designed to study the impact of land-use change on the soil microbiome, but focused on different aspects and questions. To condense the two datasets and experiments into one study, we have rewritten the Introduction and added a large section of the introduction related to land-use change to better introduce the unresolved issues in land-use change research and propose the aim and importance of our research. Note, we performed the global analysis without the data obtained in the continental scale study from China. We did this to assess whether similar conclusions were obtained when using independent datasets. The results from the continental scale study from China and the global meta-analysis complement each other and come to similar conclusions. Note that the continental scale study in China used the same methods (same soil sampling protocol, same sequencing approach) for all samples, while the global meta-analysis was based on a range of papers with different methods. As both studies came to similar conclusions, they nicely complement each other.

“Due to increasing human activities and subsequent agricultural intensification, an emerging body of research suggests that ecological communities are undergoing

fundamental changes across various spatial dimensions¹² ... agricultural intensification has also caused biotic homogenization at larger spatial scales, posing a significant concern for ecosystem services and conservation.

Although previous research revealed a decline in local diversity, local species loss is not entirely related to changes in community similarity across space¹³ ... The information about agriculture-induced biotic homogenization of belowground communities is essential for regional biodiversity planning and conservation purposes.

Land-use change and agricultural intensification can alter community assembly processes, community composition and species diversity concurrently^{13, 14, 15, 16} ... This knowledge gap hinders our comprehensive understanding of the global decline in biodiversity.

In the present study, we address three major questions: (1) whether agricultural effects lead to taxonomic and functional biotic homogenization of soil microbiomes at large spatial scales? (2) how land-use changes alter soil microbial community composition and functions across a wide range of soil and climate types, and which microbial lineages and functions are mostly impacted? ... These results will advance the holistic understanding of soil microbiome composition and function under land-use change, indicating that diversity loss at large spatial scales is one of the most substantial consequence of land-use intensification.” (L.44-107).

The major concern with this study is the direct use of functional annotation of metagenomic reads to infer changes in function. Other than bulk biogeochemistry (OM, available phosphorus, nitrogen), there are no methodological details of how the enzyme assays were conducted, so I am assuming these were inferred from the metagenomic data (KEGG, and KO annotation)? If so, this is an assumption that should be tested before inferring functional changes in the microbial community (See Rocca et al 2015 ISME, Bier et al 2015 FEMS, Hall et al 2018 Nat Microbiol). Further, linking changes in microbial genera to the functional shifts is a bit misleading, as the data are both originated from DNA, hence a correlation would be expected. Actual enzyme assays would be more robust correlations with the microbial community data, if they exist.

Response: In our study, we conducted the enzyme assays to measure the actual enzymes activities, which were not inferred from metagenomic data. We are sorry for not clearly showing methodological details of enzyme assays. We have added methodological details of enzyme assays in the section ‘Soil enzyme activities’ of Methods (L.478-489), as “The activities of soil extracellular enzymes involved in C, N and P acquisition were determined using the microplate-scale fluorometric method ¹⁷. We used a 200 μM solution of substrates labeled with 4-methylumbelliferone or 7-amino-4-methylcoumarin. The C-acquisition enzymes analyzed included β-1,4-glucosidase (BG), 1,4-β-Dcellobiohydrolase (CBH) and β-xylosidase (BX). The N-acquisition enzymes analyzed were β-1,4-N-acetylglucosaminidase (NAG) and L-leucine aminopeptidase (LAP), while the P-acquisition enzyme analyzed was alkaline phosphatase (AP). After incubation at 35 °C, plates were centrifuged, and the supernatant was transferred to black, flat-bottom 96-well plates. Fluorescence was measured using a microplate reader with 365 nm excitation and 450 nm emission filters. Soil enzyme activities were expressed as nmol g⁻¹ dry soil h⁻¹.”

Line-by-line feedback:

59 - 'Soil microbiomeS...'

Response: Done, revised as suggested.

326 - Please organize the methods as the rest of the paper is presented - continental-scale analysis, then the paired crop/natural habitat within China. Or, flip the rest of the paper to match the methods order.

Response: We have adjusted the display order of Methods to match the Results, from continental survey and sampling and global-scale meta-analysis.

339 - how was this confirmed? Please present data to indicate how similar the conditions are for the crop and adjacent natural habitat. Climate may likely be similar on ~2km scale, but water availability, etc, could easily vary across that spatial scale.

Response: Yes, we consider land-use patterns within a region to exclude the effects of

climate and soil type. The impact of land-use changes on soil microbiome is mostly attributed to changes in soil biotic and abiotic environments. We therefore measured soil biotic and abiotic indicators, including soil moisture, to assess how the response of these environmental conditions to land-use patterns affects the soil microbiome, as shown in Fig. 2 and 4. We have revised this sentence for easy understanding, “The distance between cropland and adjacent natural ecosystems is within 2 km to maintain a consistent climate and soil type.” (L.397-398).

345 - I assume the site pairs were collected within short timescales? Please confirm this timeframe.

Response: Yes, to minimize the impact of sampling times, soil samples at each site pair were collected within one day, and the whole soil survey is conducted in July and August 2019 when the crop was currently in the growing season. We have added this information in the Method, “Composite surface soil samples (top ~20 cm depth) were collected at each plot in July and August of 2019 during the crop growing season. Soil samples at each site pair were collected within one day to minimize the impact of sampling times.” (L.412-415).

394 - Please define 'available' - do you mean bio-available?

Response: Yes, available means bio-available. We have defined this in the revised MS.

426-437 (261-266) - Other than bulk biogeochemistry (OM, available phosphorus, nitrogen), there are no methodological details of how the enzyme assays were conducted, so I am assuming these were inferred from the metagenomic data (KEGG, and KO annotation)? If so, this is an assumption that should be tested before inferring functional changes in the microbial community (See Rocca et al 2015 ISME, Bier et al 2015 FEMS, Hall et al 2018 Nat Microbiol). Further, linking changes in microbial genera to the functional shifts is a bit misleading, as the data are both originated from DNA, hence a correlation would be expected. Actual enzyme assays would be more robust correlations with the microbial community data, if they exist.

Response: We are sorry for not clearly showing methodological details of enzyme assays. Enzymatic functions were measured through microplate-scale fluorometric method. We have added detailed information in the Methods in the revised MS (L.478-489).

437 - why is this abbreviation made if it is only mentioned once?

Response: Yes, we have deleted the abbreviations that only mentioned once.

715 - Don't you mean 'dissimilarities between cropland and *natural ecosystems*.'?

Response: We collected farmland and pairs of natural ecosystems, most of which were forest ecosystems. Cropland and forest cover are similar in geographic and spatial scales, so we selected forest soils as representative of natural systems to compare with croplands. We have added the illustration in the Methods, “Taking into account sample size and coverage, we selected forest to represent natural ecosystem because forest soils included more than 1,300 samples and covered six continents. Other ecosystems with only few sites or low distribution range lacked of representation for large-scale evidence (Supplementary Fig. 11), and were excluded in further analyses” (L.455-460). To avoid misunderstanding, we have revised ‘natural’ to ‘forest’ in the main text.

Figs:

Please list the actual figures *before* the extended data figs. I realize these would be properly organized in any publication, but it is confusing for the reviewers, as I saw the smaller subset of China samples before seeing the actual Figure 1.

Response: We are very sorry for the order of the figures. We would upload the main text and extended data figs separately so that the actual figures can be read before the extended data figs.

Across all meta-analysis effects, please graphically depict what is being compared (e.g. Fig 1, d, h; Fig 2 a, c; Extended Fig 7). For example, in Fig 2b/d, upper right indicates a which conversion/comparison is being made.

Response: Yes, our research focuses on forest-cropland, grassland-cropland, and wetland-cropland conversions. The results included comparison of beta diversity, relative abundance in phyla and taxa, and functional genes of these three conversions. To display the figure results more clearly, we have added textual descriptions of the effects of agricultural intensification on beta diversity, relative abundance in phyla and taxa, and functional genes in forests, grasslands, and wetlands, respectively. For the meta-analysis, we compared the microbial beta-diversity between cropland and forest ecosystems (Fig. 1b and 1c). In the text (result section), we specified Fig. 1d and h as β -diversity of both microbial taxonomic and functional composition between cropland and natural ecosystems; Fig. 2a, c and e as impacts of agriculture on the relative abundance of major phyla; Fig. 2a, c and e as impacts of agriculture on the relative abundance of major taxa.

Fig 1:

Panel c - Define 'All natural ecosystems' - meaning no cropland-conversion sites, or given the photo expansion, all four habitat categories are present at this location? Unclear what the panel C photos indicate? The cropland bracket includes all categories, but for 'All natural ecosystems' does this mean there are/are not close habitat pairings of cropland?

Panel d, h: Please indicate the sample number for each meta-analysis effect.

Response: Cropland was included in each sampling site, but not all natural systems were found and sampled in each site. All natural ecosystems means that the sampling site has all three natural ecosystems: forest, grassland, and wetland. Panel C photos show four typical ecosystem types taken during our sampling period. To be clear, we have made some modification in panel c. We have added the sample number for the analysis in panel d and h.

Extended Data Fig 1 - please further clarify this subset of points, vs. the full dataset presented in Fig 1.

Response: We replot site map to differentiate the full dataset and sub-datasets. All

sample sites were processed by amplicon sequencing, then we selected 10 sample sites to measure the metagenomic sequencing. The map is shown in Supplementary Fig 1.

Extended Fig 6 - This figure is not useful, aside from conveying that microbial genera have overlapping functional pathways. Please consider another graphical display if this is to be included.

Response: We have removed this figure and corresponding sentences in the revised MS. Figs. 1 and 3 already illustrated the functional redundancy of soil microbiomes.

Reviewer #3 (Remarks to the Author):

This study by Peng and colleagues aimed to elucidate taxonomic and functional homogenization due to intensive agriculture. The authors used a combination of meta-analysis, large-scale field study, and field trial experiments to assess the composition of soil bacterial and fungal communities. They found that agricultural land-use had a strong effect on microbial composition. While biotic homogenization is certainly an important area of research that deserves more attention, I do not think this study clearly shows that. The three studies used here are quite different from each other in terms of experimental design and questions. While they show compositional differences mediated by landuse, there is no coherent evidence that suggests homogenization. The manuscript also suffers from some critical flaws and inconsistencies. Please see my detailed comments below.

Response: Thank you for your comments and considerations! We have completely revised the manuscript based on the comments your proposed and added more content for the Introduction, Methods and Discussion in the revised MS. We have re-presented the results and some figures to focus on biotic homogenization, and provided a clearer description of the experimental purposes and focused issues of the three experiments in the revised version, making it easier for readers to understand the results and highlights of our research.

This study focuses on the question of how agricultural land use changes the composition of soil microbial communities and the functional processes they regulate. Although previous studies have explored the impact of land use changes on soil microbiomes^{18, 19}, but the conclusions were largely limited to the changes in local diversity at relative small scale. Alongside reductions in local species diversity, the consequence of land-use intensification on biotic homogenization at larger spatial scales is of great concern for ecosystem services and conservation. We still lack comprehensive understanding of the effects of agricultural land use on soil microbial composition at large scales. An important question is whether agricultural intensification caused biotic and functional homogenization of belowground communities at large scales. Although this has been carried out in animal and plant

communities for a long time, this issue has not yet been revealed in belowground communities. Thus, we used a global dataset and conducted a continental soil survey to reveal whether land use intensification leads to biotic homogenization of microbial communities at large scales. Our results demonstrate that conversion from natural ecosystems to agricultural systems leads to biotic homogenization and less variation in microbial communities (e.g. the beta-diversity across sites is significantly lower in cropland compared to grassland, wetland and forest). In addition, the continental soil survey dataset, covering different terrestrial ecosystems, was also used to explore general patterns of soil microbiome taxonomic and functional responses to agricultural intensification and how these responses vary among ecosystem types and different microbial lineages.

Overall, we combined two datasets and experiments at different scales (a global-scale meta-analysis and a continental field survey) to shed light on this ambitious and complex issue, hoping to provide more holistic and deeper insights into the ecological consequences of agricultural land use change.

Our research content and highlights are:

- 1) Revealing the impact of agricultural land use changes on the taxonomic and functional homogenization of belowground microbial communities by combining global-scale data and continental-scale survey. We assessed homogenization by determining the variation (Beta diversity) across sites for cropland versus grassland, wetland and forest. Both our global study and the continental scale study showed that there is significantly lower variation (homogenization) across croplands
- 2) Using continental-scale survey to reveal how agricultural intensification alters the composition and function of soil microbial communities by changing soil environments from various natural habitats, including forest, grassland and wetland, and obtain the results of general across ecosystems and specific-ecosystem.

Homogenization research is explored from global-scale data and continental-scale survey. Then continental-scale survey and field experiments focus on the impact of

agricultural intensification and different agricultural practices on soil microbial composition and functions. The results of these three studies are presented in the Abstract, with the biotic homogenization as a major conclusion reflected in the Title.

Introduction: the introduction is quite weak. The first two paragraphs are full of broad sentences and do not lay the foundation of this study. If biotic homogenization is the focus, I feel the introduction can be more nuanced, e.g., what are the signatures and drivers of homogenization at local, regional, and global scales? how can we identify homogenization? is it only the loss of richness? I think this would help readers understand the importance of this study. Question 2 is not novel and has been assessed by many previous studies. Were there any specific hypotheses you tested?

Response: We have re-written the Introduction to introduce relevant background and previous studies to help lay the foundation for our research as suggested. In addition, we adjust the two questions as follows: (1) whether agricultural effects lead to taxonomic and functional biotic homogenization of soil microbiomes at large spatial scales? (2) how land-use changes alter soil microbial community composition and functions across a wide range of soil and climate types, and which microbial lineages and functions are mostly impacted? We have introduced each of these two questions in the Introduction. We hypothesized that agricultural intensification causes taxonomic and functional homogenization of soil microbiomes, as is added in the Introduction.

The Introduction are following:

“Due to increasing human activities and subsequent agricultural intensification, an emerging body of research suggests that ecological communities are undergoing fundamental changes across various spatial dimensions¹² ... agricultural intensification has also caused biotic homogenization at larger spatial scales, posing a significant concern for ecosystem services and conservation.

Although previous research revealed a decline in local diversity, local species loss is not entirely related to changes in community similarity across space¹³ ... The information about agriculture-induced biotic homogenization of belowground communities is essential for regional biodiversity planning and conservation purposes.

Land-use change and agricultural intensification can alter community assembly processes, community composition and species diversity concurrently^{13,14,15,16} ... This knowledge gap hinders our comprehensive understanding of the global decline in biodiversity.

In the present study, we address three major questions: (1) whether agricultural effects lead to taxonomic and functional biotic homogenization of soil microbiomes at large spatial scales? (2) how land-use changes alter soil microbial community composition and functions across a wide range of soil and climate types, and which microbial lineages and functions are mostly impacted? ... These results will advance the holistic understanding of soil microbiome composition and function under land-use change, indicating that diversity loss at large spatial scales is one of the most substantial consequence of land-use intensification” (L.44-107).

Homogenization: I am confused regarding the homogenization indicator/metric used in this study. I am not sure if readers would agree with the main claim. Except for Figure 1b panel, the rest of the results show taxonomic and functional differences between land-use types. Sure, that information is also important, but this has been demonstrated by many studies. I was also looking for clear evidence of homogenization.

Response: Sorry for not clearly showing homogenization results in the Figures. Here, we provide large-scale evidence of agriculture-induced biotic homogenization of belowground communities from the global-scale meta-analysis and continental field survey. The meta-analysis based on dataset covering more than 2,400 soil samples across six continents showed that microbial communities were significantly more similar across space and showed higher taxonomic similarity in croplands compared to forest soils at global scale (Fig. 1b). In order to measure homogenization, we assessed the community similarity (e.g. by measuring Beta-diversity across sites). This analysis showed that croplands have much more similarity (e.g. are more homogenous). The continental field survey dataset further confirmed that the higher similarity in taxonomic and functional composition in croplands than those of natural soils (Fig. 1d and 1h, Supplementary Figs. 1 and 3). We show that the similarity across sites is higher

for cropland compared to grassland or wetland. Only, when we compared cropland versus forests, the increase was lower (but still highly significant – Fig. 1d). Taken together, our results showed widespread evidence for biotic homogenization of taxonomic and functional levels at continental and global scale, which was highlighted as a major conclusion reflecting in the Title. To be clearer, we have re-presented the results and the relevant figures (shown as following) to focus on biotic homogenization

Fig. 1. Taxonomic and functional homogenization of microbial communities in response to agricultural impacts at global (a,b) and continental scale (c-k). d, h, Effect sizes of natural ecosystems impacts on β -diversity in taxonomic composition (d) and functional composition annotated with KEGG (h) relative to croplands. Data are presented as mean \pm s.e.m. of the estimated effect sizes. Sample size is showed by number of data pairs for each group. Statistical significance is indicated by: * $P < 0.001$, ** $P < 0.01$, * $P < 0.05$.**

Supplementary Fig. 3 Response of community dissimilarity (beta diversity) to agricultural impacts between sites between cropland and forest, grassland, and wetland. (a) All 1,185 samples were included in the analysis. The total number of each site is shown in Table S2. (b) To exclude the confounding effect of sample size, five randomly selected samples from each ecosystem in a given site were included in the

analysis. Statistical significance based on Wilcoxon test.

Field experimental design: I could not understand the rationale and design of this experiment. What were the rationale and hypotheses behind imposing tillage, monoculture and fertilizer applications to forest and grassland soils? How could these results be compared with arable soils? What was the control in this experiment and how was it different from initial forests/grasslands? Were these plots in the same location? The authors say 27 in forest soils and 27 in grassland soils. What was the tillage type? Overall, this is a confusing experiment with 9 different agricultural treatments imposed on forest and grassland soils.

Response: The manuscript contains indeed a lot of data already. Below we explain our rationale. However, to focus on the main results of the meta-analysis and the continental scale analysis, we have now removed this part from the manuscript. If the editor or other reviewers feel we should keep this part in, we are happy to add it back again.

We conducted field experiment to test how different agricultural practices, such as soil tillage, crop planting and fertilizer applications, influence soil microbiomes and functions. Experimental treatments were compared to natural initial forests and

grasslands. In the process of agricultural intensification, we need to remove the vegetation on the aboveground before starting agricultural practices. Therefore, we set up a treatment: bare soil with forest/grassland vegetation removed. Soil tillage, crop planting and fertilizer applications were conducted in bare soils. The forest and grassland are at the same altitude, less than a hundred meters apart. We conducted conventional tillage (ploughing) every two months to simulate soil management.

Methods

Ln 344: A major limitation of this study is the number of plots, which varied considerably between ecosystem types. When samples are combined, this could impact the level of microbial diversity observed for each site. This could also have a confounding effect on the results. I wonder why same number of plots were not sampled for all ecosystem types. Further, the plot size is also missing. Did the plot size also vary between sites and ecosystems?

Response: In this study we compared cropland with natural ecosystems (e.g. forest, grassland and wetland). For each comparison, we made sure that we compared the same number of plots (e.g. 303 soil samples from forest and 303 soil samples from adjacent cropland). Thus, our conclusions are valid when comparing the individual natural ecosystems (e.g. forest versus cropland). Our initial plan was to collect 10 samples (10 plots) per ecosystem at each sampling site, but we need to collect samples within a limited time (July-August), so we had to make some sacrifices in the number of samples. In some remote regions, we contacted local research institutions to assist in collecting samples, resulting in possible inconsistencies in the number of samples. In addition, some samples were abandoned due to DNA extraction failure and low sequencing quality. Moreover, the number of plots of different ecosystems in a given region is basically consistent, with most sampling ecosystem having 5 or 10 plots. We have provided the number of samples for each sample site in **Supplementary Table 2**. Note, we had the lowest number of samples for grasslands (275 plots). Thus, in order to compare the same number of plots for grassland, forest (303 plots) and wetland (278 plots) we now randomly removed 28 paired plots from forest ($303-275=28$) and 3 paired

plots ($278-275=3$) from grassland. The results are similar with Fig. 1 and presented as supplementary figure 2, shown as following:

Supplementary Fig. 2 Effect sizes of natural ecosystems impacts on β -diversity in taxonomic composition with the same number of plots for grassland, forest, and wetland. Data are presented as mean \pm s.e.m. of the estimated effect sizes. Sample size is showed by number of data pairs for each group. Statistical significance is indicated by: *** $P < 0.001$, ** $P < 0.01$, * $P < 0.05$. This figure is similar to Fig. 1d with the only difference that here we standardized the number of samples analyzed to 275 per cropland-natural ecosystem comparison.

Each plot has a size of $2 \times 2 \text{ m}^2$ and is same across sites and ecosystems. Soil samples were mixed by taking three soil cores with a 5-cm-diameter auger for each plot in the surface layer. We have added the size information in the revised MS (L.414-415).

To eliminate the effect in the number of samples, we randomly selected five samples from the ecosystem of each sampling site for homogenization analysis, which was consistent with the biotic homogenization of previous results. We have added the result in the **Supplementary Fig. 3b**.

Supplementary Fig. 3 Response of community dissimilarity (beta diversity) to agricultural impacts between sites between cropland and forest, grassland, and wetland. (a) All 1,185 samples were included in the analysis. The total number of each site is shown in Table S2. (b) To exclude the confounding effect of sample size, five randomly selected samples from each ecosystem in a given site were included in the analysis. Statistical significance based on Wilcoxon test.

Supplementary Table 2. A total of 1185 soil samples including 856 paired soils across sites and ecosystems at continental scale. Number indicates the number of amplicon samples in each ecosystem in given a site. NA indicates no samples in ecosystem. A tick indicates that metagenomic data are measured in given a site.

Site	City	Cropland	Forest	Grassland	Wetland	Metagenome
1	Yulin	10	10	10	NA	
2	Shangqiu	10	10	10	10	
3	Rizhao	10	10	10	10	
4	Yinchuan	10	10	10	10	
5	Haibei	10	10	10	10	√
6	Dangxiong	10	NA	10	10	
7	Aertai	10	10	10	10	
8	Nanjing	5	5	5	5	
9	Liuan	5	5	5	5	
10	Luoyang	5	5	5	NA	
11	Zhaoqing	10	10	10	10	

12	Luding	10	10	10	10	√
13	Taigu	9	9	9	NA	
14	Zhangye	10	10	10	10	
15	Yumeng	5	4	5	5	
16	Songyuan1	10	10	10	NA	
17	Guiyang	10	10	10	10	
18	Guangzhou	5	NA	5	5	
19	Yueyang	5	5	5	5	
20	Linzhi	10	10	10	10	√
21	Liuba	7	7	7	7	√
22	Wujiaqu	5	5	NA	NA	
23	Taibai	10	10	10	10	
24	Kuerle	10	10	10	10	√
25	Songyuan2	5	5	5	5	
26	Changchun	5	5	NA	5	
27	Xuancheng	5	5	5	5	
28	Daqing	5	5	5	5	
29	Haerbin	5	5	NA	5	
30	Suihua	5	5	5	5	√
31	Chengde	5	5	5	5	√
32	Chaozhou	5	5	5	5	
33	Songming	6	6	5	5	
34	Fuzhou	5	5	5	4	
35	Xinyang	8	8	8	8	√
36	Tonghua	8	8	NA	8	
37	Jiuquan	10	10	NA	NA	
38	Yichang	10	10	NA	10	
39	Langzhong	5	5	5	5	
40	Kaihua	10	10	10	10	√
41	Wuzhou	5	5	3	5	√
42	Tongzi	5	5	3	5	
43	Haikou	6	6	NA	6	
44	Lasa	10	NA	10	10	

Number of soil samples: how did you come up with 1185 samples (Ln 354)? Abstract says 856 paired soils. The supplementary Information section needs more development. Figure captions are inadequate.

Response: We collected 1,185 samples covering four ecosystems at continental scale. At each site, agricultural soils were sampled, but parts of the natural systems were not. The number of samples for each sample site were provided in Supplementary Table 2 and sample information were added in the Methods. In sum, we obtained a total of 303 forest-cropland pairs, 275 grassland-cropland pairs, and 278 wetland-cropland pairs. We have added the sample number for effect analysis in Figure and Figure captions.

Fungal ASVs: The authors amplified the ITS region to assess fungal ASVs. Due to the inherent variability of ITS region, as shown in many recent studies, I do not think ITS data can be used to obtain amplicon sequence variants (ASVs). Even variation of a single base might produce a new ASV. I am not sure if fungal ASV data are admissible. There are also some inconsistencies and missing information. What was the total number of ASVs for 16S and ITS?

Response: Yes, we acknowledge that there are inherent limitations associated with the use of an ITS region for enabling in-depth characterization of fungal communities despite improvements in sequencing techniques and bioinformatics approaches. However, the ITS region is by far the best option as a general DNA (meta) barcoding marker for fungi, and is still widely used as the standard in fungal community research. In our research, we took the fungal communities as one of the driving factors affecting the bacterial communities to reveal the underlying mechanism of the impact of land-use change on bacterial composition. We were not concerned with changes at the fungal species level, so ITS region sequencing should have limited impact on our research and results. We have added some descriptions to noted this point, as “It should be noted that although the ITS region is by far the best option as a general DNA (meta) barcoding marker for fungi, there are inherent limitations associated with the use of a ITS region for enabling in-depth characterization of fungal communities. We were not concerned with changes at the fungal species level, so ITS region sequencing should have limited

impact on our results” (L.519-524).

In the present study, a total of 31,402 bacterial ASVs and 77,962 fungal ASVs were remained for further analysis. We have added detailed information in the Methods (L.524-526).

Amplicon and metagenomic sequencing: This section is missing a lot of relevant details. The authors should add information related to various steps (e.g., library preparation, trimming, merging,). What kind of MiSeq sequencing did you perform (250, 300 paired end)? What were the controls? For metagenomics, how were the 10 samples chosen for each ecosystem type?

Response: We have added detailed information in amplicon and metagenomic sequencing in the revised MS. The diversity of soil bacteria and fungi was measured by 16S rRNA gene and nuclear ribosomal ITS amplicon sequencing using an Illumina MiSeq PE250 platform. Metagenomic sequencing were performed on the Illumina Illumina Novaseq 6000 platform to generate 150 bp paired-end reads. For the selection of 10 sampling sites for metagenomic sequencing, firstly, each site should include all ecosystem types, including cropland, forest, grassland and wetland; secondly, the spatial range covered by these 10 sites is similar to that of all sampling sites to present general patterns. The sites for amplicon and metagenomic sequencing were shown in **Supplementary Fig. 1a**.

The revised methods were following:

“DNA extraction, amplicon sequencing and data preprocessing

Genomic DNA was extracted from 0.5 g of the soils using the MP FastDNA spin kit for soil (MP Biomedicals, Solon, OH, USA) according to the manufacturer’s instructions. The diversity of soil bacteria and fungi was measured by 16S rRNA gene and nuclear ribosomal ITS amplicon sequencing using an Illumina MiSeq PE250 platform. For bacterial community, 16S rRNA genes were amplified using primer set 515F and 907R, targeting the V4-V5 region of the 16S rRNA gene. For fungal community, the first nuclear ribosomal ITS sequences were amplified using primers ITS5-1737F/ITS2-2043R, targeting the ITS1-5F region. PCR amplification was

performed in a 50 µl volume: 25 µl 2x Premix Taq (Takara Biotechnology, Dalian Co. Ltd., China), 1 µl each primer (10 µM) and 3 µl DNA (20 ng/µl) template. The PCR thermal cycling conditions were performed by thermocycling: 5 min at 94 °C for initialization, followed by 30 cycles of 30 s denaturation at 94 °C, 30 s annealing at 52 °C, 30 s extension at 72 °C, and 10 min final elongation at 72 °C. The length and concentration of the PCR product were detected by 1% agarose gel electrophoresis. Sequencing libraries were generated using NEBNext® Ultra™ II DNA Library Prep Kit for Illumina® (New England Biolabs, MA, USA) following manufacturer's recommendations and index codes were added. Bioinformatic processing, including filtering, dereplication, sample inference, chimera identification, and merging of paired-end reads, was performed using the Divisive Amplicon Denoising Algorithm 2 (DADA2) package in R²⁰. In brief, the plotQualityProfile command was run to detect the quality of the amplified sequences. We imposed a minimum length of 100 bp to remove any small fragments at the filtering stage, at which, the error in the maxEE argument was 2 as this optimised the retention of reads throughout the pipeline. Error rates were subsequently calculated by the DADA2 algorithm before dereplication and merging of paired end sequences. Chimeras were removed using the removeBimeraDenovo command with method = "consensus"²¹. Finally, the taxonomical annotation of the representative sequences of amplicon sequence variants (ASVs) was performed with a naïve Bayesian classifier using the Silva v. 138 (for bacteria) and the UNITE v. 7 (for fungi) database^{22,23}. It should be noted that although the ITS region is by far the best option as a general DNA (meta) barcoding marker for fungi, there are inherent limitations associated with the use of a ITS region for enabling in-depth characterization of fungal communities. We were not concerned with changes at the fungal species level, so ITS region sequencing should have limited impact on our research and results. The sequence number in each sample was rarefied to the same depth for the 16S rRNA gene (15,000 reads) or ITS sequences (21,921 reads), leaving a total of 31,402 bacterial ASVs and 77,962 fungal ASVs for further analyses.

Shotgun metagenome sequencing

A subset of 40 samples from 10 regions covering cropland, forest, grassland and

wetland soils was selected for metagenomic sequencing to analyze changes in microbial community functional potential (n = 10 per ecosystem type; Supplementary Fig. 1a). Metagenomic libraries for 40 samples were prepared according to the product instructions of ALFA-SEQ DNA Library Prep Kit (Findrop, Guangzhou, China) and index code was added. Initial quantification of the library concentration was performed using Qubit 3.0 fluorometer (Life Technologies, Carlsbad, CA, USA) and the library was diluted to 1 ng/μL. Agilent 2100 Bioanalyzer System (Agilent Technologies, CA, USA) was used to detect the integrity of library fragments and the length of insert size. Then, the library was sequenced on Illumina Novaseq 6000 platform (Illumina, San Diego, CA, USA) to generate 150 bp paired-end reads at Guangdong Magigene Biotechnology Co., Ltd. In total, 1.71×10^9 raw reads were sequenced across all samples, which yielded 512.3 Gbp of total sequence information with an average data volume of 12.8 Gbp per sample. Raw data were quality checked with FastQC (v0.11.9) and processed using Trimmomatic v.0.39 (leading: 3, trailing: 3, slidingwindow: 4:15, minlen:36) to trim adapters and discard bases with a quality score <15 and length < 36 bp. After that, 12.2 Gbp clean data per sample were obtained. Clean reads were annotated for functional analysis of the microbiome using HUMAnN v3.7 (based on DIAMOND (version 2.1.6) ²⁴ and Bowtie2 (version 2.5.1) ²⁵) with ChocoPhlAn database (version “mpa_vJan21_CHOCOPhlanSGB_202103”) and UniRef90 (version “uniref90_201901b”) protein database to quantify relative abundance of functional genes and metabolic pathways ²⁶. The annotation results were organized according to Kyoto Encyclopedia of Genes and Genomes (KEGG) Orthologues (KOs), Clusters of Orthologous Group of proteins (COG) functional categories and MetaCyc functional pathways using “humann3_regroup_table” script. The abundance of functional gene was expressed as Transcripts per million” (L.490-553).

Statistical analyses: The analysis performed in this study is unbalanced and under-described. For functional annotation, they used fungal ITS. However, for statistical analyses (e.g., beta-diversity, SEM), they only used bacterial 16S. This is odd! Also, please add the number of samples for important analyses. For example, what was the

total number of samples for SEM?

Response: We are sorry for the misunderstanding caused by not clearly presenting our research. Our three data and experiments are all about soil bacteria. Our study did not include the effects of agricultural land use change on soil fungal communities. We only took the fungal communities as one of the driving factors affecting the bacterial communities to reveal the underlying mechanism of the impact of land-use change on bacterial community composition.

We have added the sample number for effect analysis in the Methods and Figure captions. SEM were conducted in 40 site samples including amplicon and metagenomic sequencing datasets.

Meta-analysis: Ln 507- what proportion of samples were sequenced using 515F and 806R? Ln 511- what was the quality threshold? So, out of 3482 samples, 1300 were forest soil samples? I see that other ecosystems were removed from further analyses. So, it is important they call it forest and not ‘natural ecosystems’.

Response: We have added detailed information in the meta-analysis. After the initial selection, 75 studies were left, over 6,000 sample sequencing data. In these studies, high-throughput sequencing of bacterial communities was conducted using Illumina, Ion S5, and 454 pyrosequencing platforms. Twenty-three primer pairs were identified from the research metadata, and the most used primers in the sample were 515F and 907R (18/75), 515F and 806R (13/75), and 338F and 806R (16/75). We utilized the `-fastq_filter` command in the `vsearch` tool for sequence quality control, with the parameter `-fastq_maxee` set to 1. This implies that the maximum expected errors (EE) threshold for low-quality bases in all sequences is set to 1. Only sequences with an expected error count less than or equal to 1 are retained, while sequences exceeding this threshold are filtered out. Of the 3,482 samples analyzed, 1,300 samples were from forest soils. To avoid misunderstanding, we have revised ‘natural’ to ‘forest’ in the main text. In total, 2,403 samples were included in the global-scale meta-analysis. We have added the illustration in the Methods (L.455-460). PRISMA flow diagram for meta-analysis were added in the Supplementary Fig. 10.

Supplementary Fig. 10 PRISMA flow diagram for the studies selected and included in the systematic review.

Methods shows they only used FungalTraits, but then I also see FAPROTAX. Which samples were annotated with FAPROTAX?

Response: We have added FAPROTAX database in the Methods (L.xxx). FAPROTAX analysis was conducted using 1,185 samples to test the effects of

agricultural intensification on abundance of major nutrient cycling groups at continental scale. We added this information in Fig. 2 legend.

Results

Figures: Finding the main figures was quite difficult. There is a whole extended data section between figure legends and the main figures, so it was challenging to understand the figures and source of samples. As there are three studies in this study, figures should be clearly marked, i.e., results from which study they are showing specifically. It is important that authors describe figures categorically and not combine everything, as the three studies are very different in terms of design and settings.

Response: We are very sorry for hard understanding. This may be because the formatting order was disrupted when merging the files. We would upload the main text and extended data figs separately so that the actual figures can be read before the extended data figs.

We have revised the Results for describing figures clearly and divided the results into 4 sub-headings: (1) Agriculture causes biotic homogenization of taxonomic and functional profiles; (2) Agricultural effects on specific microbial phylotypes and functions; (3) Mechanisms underlying changed microbial communities; (4) Links between microbial communities and soil functions; The first part, related to homogenization, contains global meta-analysis and continental-scale survey datasets. Sections 2, 3 and 4 use continental-scale survey dataset. Section 5 shows the results of field experiments. Moreover, we have revised the figure legends and indicated the sample source: (1) Fig. 1. Taxonomic and functional homogenization of microbial communities in response to agricultural impacts at global and continental scale; (2) Fig. 2. Effects of agricultural land use on different microbial taxa at continental scale; (3) Fig. 3 Agricultural effects on functional genes involved in biogeochemical cycling processes at continental scale; (4) Fig. 4 Environmental drivers of microbial composition and their relationship to soil functions at continental scale.

Discussion

There is hardly any discussion! It is similar to the abstract in terms of length. For a manuscript with three large studies, this section is insufficient and should be rewritten.

Response: Thanks for your comments. We have re-written the Discussion section to discuss the three studies in detail.

In the 1st paragraph, we summarized the main findings of this study, as in L249-260: “Agricultural land-use change has exerted profound effects on belowground biodiversity^{19,27}, and the effects are likely to accelerate in the coming decades²⁸ ... The variation across sites (Beta-diversity) was significantly lower across croplands compared to grasslands, wetlands and forests, pointing to biotic homogenization in croplands.

In the 2nd-4th paragraph, we mainly discussed the taxonomic and functional homogenization of soil microbiomes following agricultural intensifications, and highlighted its implication for developing agricultural management strategies to conserve landscape-scale biodiversity and ecosystem service provision, as in L261-310: “Although land-use changes and agricultural intensification have been proven to be major drivers to biodiversity loss^{29, 30}, positive impacts of agriculture are often observed at regional and local scales ... Overall, our study provides a comprehensive insight that agricultural land-use change cause biotic homogenization in taxonomic and functional composition, and has major implications for developing agricultural management strategies to conserve landscape-scale biodiversity and ecosystem service provision^{31,32}.

Biotic homogenization in response to agricultural impacts is a multifaceted process that involves considering the invasion and extinction of species, as well as the heterogeneity of landscapes ... Frequently disturbed soil environments can promote the gains and proliferation of novel species and the gradual replacement of locally distinct communities by cosmopolitan communities via altered competitive and coexistence dynamics³³, homogenizing assemblage composition across space.

The influence of agricultural intensification on biotic homogenization might be attributed to the reduction in environmental heterogeneity in monoculture-dominated landscapes³⁴. Landscape heterogeneity is central to the spatial organization of

ecological communities ³⁵ ... Given the complexity of the soil environment, more attention needs to be paid to the biotic homogenization caused by agricultural intensification of the soil microbiome at various scales.”

In the 5th-6th paragraph, we explained how land-use changes alter soil microbial community composition and functions, and which microbial lineages and functions are mostly impacted, as in L335-379: “Our results showed that land use change had a greater impact on taxonomic composition than on functional composition, indicating the functional redundancy of soil microbiomes ^{10, 36, 37} ... This confirms the notion that there is a weak association between stable functional composition and variable taxonomic composition in microbial systems ³⁶.

Changes in soil microbial communities across space are often strongly correlated with differences in soil abiotic and biotic conditions ³⁸ ... Many taxa with spore-forming ability had a higher species pool effect, indicating that these taxa survived and outcompeted environmental stress and were retained during land use changes or were more likely to spread from natural ecosystems due to their own adaptive capabilities.”

In the last paragraph, we draw a conclusion of our study and made some outlooks, as in L380-389: “Our findings have significant implications for predicting ecological consequences of land-use change and agricultural management. The links between microbial composition and ecosystem function suggest that biotic homogenization have previously unrecognized and negative consequences for agricultural sustainability and service ... Our study suggests that biotic homogenization of belowground microbiome across large spatial scale should be taken into account when evaluating the sustainability, soil health and crop performance of agricultural management practices”.

References:

1. Wang H, *et al.* Large-scale homogenization of soil bacterial communities in response to agricultural practices in paddy fields, China. *Soil Biol Biochem* **164**, 108490 (2022).
2. Haq IU, Zhang M, Yang P, van Elsas JD. Chapter Five - The Interactions of Bacteria with Fungi in Soil: Emerging Concepts. In: *Advances in Applied Microbiology* (eds Sariaslani S, Gadd GM). Academic Press (2014).
3. Boer Wd, Folman LB, Summerbell RC, Boddy L. Living in a fungal world: impact of fungi on soil bacterial niche development. *FEMS Microbiology Reviews* **29**, 795-811 (2005).
4. Stegen JC, *et al.* Quantifying community assembly processes and identifying features that impose them. *ISME J* **7**, 2069-2079 (2013).
5. Jiao S, Yang Y, Xu Y, Zhang J, Lu Y. Balance between community assembly processes mediates species coexistence in agricultural soil microbiomes across eastern China. *ISME J*, (2019).
6. Oksanen J, *et al.* Package ‘vegan’. *Community ecology package, version 2*, 1-295 (2013).
7. Bates D, Mächler M, Bolker B, Walker S. Fitting Linear Mixed-Effects Models Using lme4. *J Stat Softw* **67**, 48 (2015).
8. Love MI, Huber W, Anders S. Moderated estimation of fold change and dispersion for RNA-seq data with DESeq2. *Genome Biology* **15**, 550 (2014).
9. Rosseel Y. lavaan: An R Package for Structural Equation Modeling. *J Stat Softw* **48**, 36 (2012).
10. Louca S, Parfrey LW, Doebeli M. Decoupling function and taxonomy in the global ocean microbiome. *Science* **353**, 1272-1277 (2016).
11. Pölmel S, *et al.* FungalTraits: a user-friendly traits database of fungi and fungus-like stramenopiles. *Fungal Diversity*, (2021).
12. Benton TG, Bieg C, Harwatt H, Pudasaini R, Wellesley L. Food system impacts on biodiversity loss. *Three levers for food system transformation in support of nature Chatham House, London*, 02-03 (2021).
13. Gossner MM, *et al.* Land-use intensification causes multitrophic homogenization of grassland communities. *Nature* **540**, 266-269 (2016).

14. Felipe-Lucia MR, *et al.* Land-use intensity alters networks between biodiversity, ecosystem functions, and services. *Proceedings of the National Academy of Sciences* **117**, 28140-28149 (2020).
15. Rodrigues JLM, *et al.* Conversion of the Amazon rainforest to agriculture results in biotic homogenization of soil bacterial communities. *Proceedings of the National Academy of Sciences* **110**, 988-993 (2013).
16. Cornell CR, *et al.* Land use conversion increases network complexity and stability of soil microbial communities in a temperate grassland. *ISME J*, (2023).
17. Bell CW, Fricks BE, Rocca JD, Steinweg JM, McMahon SK, Wallenstein MD. High-throughput fluorometric measurement of potential soil extracellular enzyme activities. *JoVE (Journal of Visualized Experiments)*, e50961 (2013).
18. Banerjee S, *et al.* Agricultural intensification reduces microbial network complexity and the abundance of keystone taxa in roots. *ISME J* **13**, 1722-1736 (2019).
19. Tsiafouli MA, *et al.* Intensive agriculture reduces soil biodiversity across Europe. *Glob Change Biol* **21**, 973-985 (2015).
20. Callahan BJ, McMurdie PJ, Rosen MJ, Han AW, Johnson AJA, Holmes SP. DADA2: High-resolution sample inference from Illumina amplicon data. *Nat Methods* **13**, 581-583 (2016).
21. Modin O, *et al.* Hill-based dissimilarity indices and null models for analysis of microbial community assembly. *Microbiome* **8**, 1-16 (2020).
22. Quast C, *et al.* The SILVA ribosomal RNA gene database project: improved data processing and web-based tools. *Nucleic Acids Res* **41**, D590-D596 (2013).
23. Yilmaz P, *et al.* The SILVA and “All-species Living Tree Project (LTP)” taxonomic frameworks. *Nucleic Acids Res* **42**, D643-D648 (2014).
24. Buchfink B, Xie C, Huson DH. Fast and sensitive protein alignment using DIAMOND. *Nat Methods* **12**, 59-60 (2015).
25. Langmead B, Salzberg SL. Fast gapped-read alignment with Bowtie 2. *Nat Methods* **9**, 357-359 (2012).
26. Beghini F, *et al.* Integrating taxonomic, functional, and strain-level profiling of

- diverse microbial communities with bioBakery 3. *eLife* **10**, e65088 (2021).
27. Wall DH, Nielsen UN, Six J. Soil biodiversity and human health. *Nature* **528**, 69-76 (2015).
 28. Newbold T, *et al.* Climate and land-use change homogenise terrestrial biodiversity, with consequences for ecosystem functioning and human well-being. *Emerging Topics in Life Sciences* **3**, 207-219 (2019).
 29. Foley Jonathan A, *et al.* Global Consequences of Land Use. *Science* **309**, 570-574 (2005).
 30. Maxwell SL, Fuller RA, Brooks TM, Watson JEJNN. Biodiversity: The ravages of guns, nets and bulldozers. *Nature* **536**, 143 (2016).
 31. van der Plas F, *et al.* Biotic homogenization can decrease landscape-scale forest multifunctionality. *Proc Natl Acad Sci USA* **113**, 3557-3562 (2016).
 32. Le Provost G, *et al.* The supply of multiple ecosystem services requires biodiversity across spatial scales. *Nat Ecol Evol*, (2022).
 33. Ladouceur E, *et al.* Linking changes in species composition and biomass in a globally distributed grassland experiment. *Ecol Lett* **n/a**, (2022).
 34. Fahrig L, *et al.* Functional landscape heterogeneity and animal biodiversity in agricultural landscapes. *Ecol Lett* **14**, 101-112 (2011).
 35. Montoya-Sánchez V, *et al.* Landscape heterogeneity and soil biota are central to multi-taxa diversity for oil palm landscape restoration. *Communications Earth & Environment* **4**, 209 (2023).
 36. Louca S, *et al.* Function and functional redundancy in microbial systems. *Nat Ecol Evol* **2**, 936-943 (2018).
 37. Louca S, *et al.* High taxonomic variability despite stable functional structure across microbial communities. *Nat Ecol Evol* **1**, (2017).
 38. Labouyrie M, *et al.* Patterns in soil microbial diversity across Europe. *Nat Commun* **14**, 3311 (2023).

Reviewers' Comments:

Reviewer #1:

Remarks to the Author:

I think the authors have satisfactorily addressed the concerns in the previous version of this manuscript. Overall, I am supportive of publication of this work.

Minor comments:

The authors used both asterisks and P values in the same figure, such as Figure 1. Please try to use the method.

For the Abstract, can the authors further highlight how bacterial composition were altered, especially in Line 34-35. It is not clear how '41-45% of phylotypes' altered? Are the abundances of these phylotypes decreased/lost or increased?

Reviewer #2:

Remarks to the Author:

Thanks for the comprehensive responses and corresponding amendments to the manuscript, per my feedback and the valuable feedback of the other reviewers.

Glad you were able to clarify that enzyme activity wasn't simply inferred from the metagenomic data, and instead standard fluorometric assays measured enz max capacity.

Reviewer #4:

Remarks to the Author:

Hello authors,

This is a very nice study with big implications given the scale of the analyses, and the importance of soil bacterial diversity and functionality undergoing massive land-use changes globally. Also, I should say that after reading previous reviewer comments, it appears that this manuscript is much more well organized and substantive than previous version/s, and generally conveys the scientific goals and questions well. That said, I think there are two things that need some attention, namely the word choice and general writing flow (grammar) and the Discussion section. I think some word choices are either not translating very well or, in my opinion, simply the wrong word used. For example, the authors repeatedly refer to 'agricultural intensification.' To me (and likely the reader), this implies that they are talking about different scales of agricultural management 'intensity', typically from sustainable-to-conventional (i.e. no-till, low chemical use, mixed crop to tillage, high chemical input, monoculture), which would be incredibly interesting, but was not the focus of this paper, rather they looked at land conversion to some form of agriculture. Secondly, the Discussion still needs quite a bit of work, particularly, the authors need to 1) focus their discussion of other (prior) studies and provide substantive examples that link or contradict others' results, and 2) Align topic sentences with the entire paragraphs. While some of the topic sentences are interesting and well-worded, many times they don't actually fit the major point provided in the following sentences, therefore this requires some editing. Finally, I feel that while the figures (namely the trees in Figure 2) are all beautiful, there are too many (sub-figures) and some of the time, they do not convey enough (simply-interpretable) information for the reader. For example, it is almost impossible to determine what I should take away from the very large phylogenetic tree figures (Figs 2B,D,F). Therefore I suggest reviewing which figures you believe convey the most important information to the reader, and move the extras to supplemental files (see my more detailed comments regarding figures).

Large-scale evidence of agriculture-induced taxonomic and functional homogenization of belowground communities

Summary

This study analyzes continental (China) and global (meta-analysis) trends in land use conversion effects from natural ecosystems (e.g. wetland, grassland, forest) to agricultural production systems on soil bacterial communities and functions. They use a suite of methods, including amplicon sequencing, metagenomics and statistical analyses (e.g. FAPROTAX) to calculate function, and compare the changes across natural systems to determine if there are broad global consistencies. They show that agricultural conversion leads to more homogeneous bacterial communities (mostly relative abundance-based taxonomic similarity) both globally and continentally, and that these effects could have impacts on long-term soil bacterial community functions, particularly for N and P (less so for C metabolism functions).

General Comments

This is a very nice study with big implications given the scale of the analyses, and the importance of soil bacterial diversity and functionality undergoing massive land-use changes globally. Also, I should say that after reading previous reviewer comments, it appears that this manuscript is much more well organized and substantive than previous version/s, and generally conveys the scientific goals and questions well. That said, I think there are two things that need some attention, namely the word choice and general writing flow (grammar) and the Discussion section. I think some word choices are either not translating very well or, in my opinion, simply the wrong word used. For example, the authors repeatedly refer to 'agricultural intensification.' To me (and likely the reader), this implies that they are talking about different scales of agricultural management 'intensity', typically from sustainable-to-conventional (i.e. no-till, low chemical use, mixed crop to tillage, high chemical input, monoculture), which would be incredibly interesting, but was not the focus of this paper, rather they looked at land **conversion** to some form of agriculture. Secondly, the Discussion still needs quite a bit of work, particularly, the authors need to 1) focus their discussion of other (prior) studies and provide substantive examples that link or contradict others' results, and 2) Align topic sentences with the entire paragraphs. While some of the topic sentences are interesting and well-worded, many times they don't actually fit the major point provided in the following sentences, therefore this requires some editing. Finally, I feel that while the figures (namely the trees in Figure 2) are all beautiful, there are too many (sub-figures) and some of the time, they do not convey enough (simply-interpretable) information for the reader. For example, it is almost impossible to determine what I should take away from the very large phylogenetic tree figures (Figs 2B,D,F). Therefore I suggest reviewing which figures you believe convey the most important information to the reader, and move the extras to supplemental files (see my more detailed comments regarding figures).

Specific Comments

Figures

Figure 1 h + j - I do not understand how J would indicate such similarity while an effect size of KEGG betadiversity (based on Bray-Curtis?) and a PCoA of Bray-Curtis show such different results...?? This intuitively does not make sense given they appear to be using the same data and show very different results for grassland vs cropland KEGG function. For example, it would appear that the forest -> cropland functional comparison in Figure i would have stronger and likely significantly different functional results for the 'effect size', which we do not see in Figure h. This leaves me with questions rather than answers, please confirm data analysis/results are correct.

Figure 2. While beautiful, it is very hard to discern the MOST important things to take away from these figures. The trees are beautiful, but it's very hard to gather the results. Perhaps keep A,C,E, (which have all the taxonomic and functional responses) and move B,D,F to supplement...? Also, in A,C,E please change the figure titles, as 'Agricultural effect on abundance in xx' does not make too much sense. Something like "Forest taxonomic & functional abundance relative to croplands". Also, in A,C,E please change the green descriptor (y axis) from 'Nutrient cycling' to 'Functionality' or "Function" as they are not all nutrient related responses...

Figure 3. D - I suggest you keep blue as negative and red as positive to match other figures (e.g. Fig4 correlations and the SEM)

Figure 4B - this is a very confusing figure to people that have not worked with this analysis, and to be direct, it tells me nothing unless you can create some sort of formatting that indicates what the differences in sites are, or if you can split the sites by altered community assembly processes on one side and unaltered assembly processes on the other to make it more visually easier to understand the point. Overall, 4C is much easier to understand and I suggest moving 4B to the supplement.

Fig 4E - You barely discuss this in the results or discussion section, and it is largely, in my opinion, an unimportant result. A consistently negative correlation between one principal coordinate axis of bacterial B-C dissimilarity and microbial functions in all ecosystem types is not terribly interesting, unless you present something you would like to discuss further, for example: more similar (less diverse) bacterial communities expressing more functions than less similar (more diverse) bacterial communities... which is slightly counterintuitive and could be interesting. But, given how many results you already have, I suggest moving 4E to the supplement.

Manuscript

L24: can just say aboveground

L33: "loss of soil bacteria with restricted ranges (endemic)" is a very specific conclusion regarding dispersal capacity, niche breadth, etc... I think this terminology may have to be proven more definitively to say this. Additionally, it does not make too much sense that the 'loss' of taxa with 'restricted ranges' would lead to homogenization. I suggest rewriting: "Our results

demonstrate large-scale bacterial taxonomic and functional homogenization upon conversion to agriculture, mainly driven by the loss of soil bacteria.”

L39: I would suggest adding: “reduction in functional diversity *provided by soil bacteria* in cropland soil.”

L31-36: I suggest moving sentences from L33-36 (Agricultural...*Latescibacterota*) above the previous sentence (Our results...)

L49: revise: have focused on local species diversity ...

L50-51: ...are relevant to highlight the loss of global biodiversity...

L84: isn't it really agricultural *conversion* rather than agricultural intensification here ??

L86: i would suggest adding: ...biodiversity and associated ecosystem functions.

L91: most *strongly* impacted?

L101: For *the* second question,

L106: land-use change, and demonstrate ... scales is **a** consequence of land-use intensification *through agricultural conversion*.

L115: cropland soils are more similar **than paired natural ecosystem soils**.

L125: loss of habitat-specific taxa

L127-132: This section is a stretch for me. I would not consider there being slightly more samples from croplands than the original habitat type that harbor shared taxa (ubiquitous) being a range expansion. In fact, because in many cases these samples would be from land conversions e.g. from forest to agriculture, therefore the 'range' is just a different habitat type. Secondly, on L131-132 'loss of habitat-specific endemics in croplands' is a little confusing as well... especially when considering SFig6A (>50% of taxa are unchanged when converted to ag, and fairly similar proportions that are 'enriched' or 'deleted'). Croplands surely also have their own 'unique' taxa that are not shown in the figures, so yes, the community changes (richness and beta diversity) and the taxa that are unique ('endemic') to e.g. grasslands are lost, but 1) there are many more shared taxa than unique taxa and 2) there are also habitat-specific endemics in croplands.

My issue with this phrasing trying to connect bacteria taxa to endemism or range expansion is that it assumes a lot of functional/physiological specificity for bacteria based on sequencing data from soil. It's not that I don't believe it might be true, but it feels like an unnecessary logical leap to concepts that are almost untestable in reality (thousands of taxa that can't be cultured). So

my suggestion is to re-write and clarify this section or to simply present the results without suggesting range expansion or loss of 'endemics' (which I don't think is a term that should be used for taxa unique to particular soil sample's habitat type anyway.)

In my opinion, homogenization implies lots of bad things, including the simplification/loss of really important functional capacities (associated with diversity), and so these specific definitions of 'biotic homogenization' you have used may be unnecessary and this section can be stated more simply. I suggest this paper for some other context regarding homogenization etc: Kortz & Magurran 2019 - <https://royalsocietypublishing.org/doi/10.1098/rsbl.2019.0133>

L176: observed in **the** metagenomic dataset

L177-178: what is meant by 'categories' ? Functional classifications from what analysis metagenomics mapped to KEGG??

Edit: functional categories found in croplands exhibited consistent changes in direction when compared to the three natural ecosystems after aggregating all *metagenomic*? functional category *count data*? from *KEGG/FAPROTAX*? above level three (*I'm not sure what the data are here?* specify what data and what analysis this is referring to).

L181: what are degradative genes (carbon-degrading?), please be more specific.

L182: when you say 'agricultural *intensification*' I think you should be saying instead **agricultural conversion**. As you do not have treatment/s or treatment gradients for these analyses (for example different levels of fertilization, plowing, pesticide application, etc which would be associated with ag intensification).

L188: make this a new paragraph at **Agriculture significantly...**

L189: a number of **functionally important** N cycling, P utilization, and sulfur metabolism **genes**

L193: application of fertilizers **and/or the loss of leguminous plant taxa found in natural ecosystems**. (important to also consider the loss of important plant taxa in ag that affect microbial functions such as nifH as well)

L206-206: I like when you introduce the ecological importance/role of these gene/trait responses, but I think you should elaborate on this resuscitation-promoting gene here. Use a citation that highlights that this means and why this makes sense in croplands versus e.g. a forest

L209: I think you should specify how community assembly was calculated when presenting this result. For example: ...dominated microbial community assembly (**as calculated using ...analysis**) in croplands...

L211: At the same time, agriculture, acting as... (delete 'impacts')

L211: filter, continues

L213: again, I'm not sure if 'intensification' is the word you want here. This point regarding agricultural management leading to more homogenous abiotic and biotic conditions is nice, but agricultural management does also imply a lot of seasonal disturbances (e.g. fertilization, chemical additions, OM additions, plowing, etc.), so while they may be consistent disturbances I think you want to avoid the word 'intensification' as it suggests management practices will be progressively more disruptive, which then runs counter to the concept of abiotic/biotic homogenization for the soil bacterial tax. Please keep an eye on this word throughout the manuscript and be sure you actually mean intensification and not something like conventional management practices (such as plowing, pesticide and fungicide applications, etc.)

L226: composition or abundance??

L238: what is meant by a 'coupled relationship'? Just be more explicit with what you mean here, because as written it just seems like the metagenomic data and the enzyme data did not reveal the same results (actually suggesting a potential problem with methods being able to consistently resolve microbial functioning in soil).

L255: different agricultural practices?? I didn't see results differentiating the type of agricultural management/cropping system, etc?

L256: of taxonomic and, **to a lesser degree**, functional homogenization

L257: delete 'landscapes'

L258: **taxonomic** variation

L259: significantly lower **in** croplands **than in** grasslands, ...

L263-266: I don't really understand why the positive effects of agriculture on aboveground diversity (richness?) are being discussed here? Including the relative increases in 'exotic' species?? This is just kind of strange, and doesn't fit. I suggest deleting this part and getting from a better topic sentence to the relationships observed where alpha metrics can increase at the expense of betadiv.

L270: ... animal groups **both** above- and below-ground

L277: I think it is important to remind the reader that the functional results (though significant in some cases) were not as strong as the taxonomic results. This is still a very important finding, but you should simply state something like: "Although taxonomic homogenization in cropland versus natural ecosystems was stronger and more significant in many cases, we observed very important microbial functional shifts under croplands, including functional homogenization."

L283-286: I'm not really sure what the purpose of this statement is, and also it doesn't make much sense to me. How would massive amounts of fertilizer inputs "likely not only to be the cause for reduced functional diversity." ??? Perhaps this is just a grammatical error, or you are trying to make a different point, but you should edit this sentence. You then go on to state, "external inputs may also compensate for reduced functional diversity and functioning." As written, it appears you are suggesting that we can just use agricultural chemicals (that are often bad for the environment) to offset the losses of microbial functioning in soil? I think this statement should be either removed or revised to be more explicit about your meaning.

L288: I think more detail is needed for these two studies you cite, for example taxonomic homogenization was identified for EU grasslands compared to agricultural intensification? Or was there just some sort of management of the grassland that homogenized taxa/functions? Readers would not understand exactly the connection between your analysis and the two papers cited.

L291: what are the major implications? That we should stop converting land to agriculture? If so, say that. If you mean some more complex land use management strategy/national regulations with enforcement plans, suggest that. As written, this is kind of an empty statement.

L298: I don't know what this means??: cause some general effects on soil communities ??
Rewrite this

L301: result in the **trait-based** filtering out of certain species, leading to...

L303: what do you mean by habitat filtering?? Provide a brief example to allow the reader to understand how abundant colonizer species would benefit and 'win' compared to a specialist.

L305-306: similarly, why wouldn't there just be new specialist taxa in the new agriculture habitat? There would be some 'loser' specialists and some 'winner' specialists according to this logic

L308-309: why would cosmopolitan communities be able to outcompete locally distinct communities after frequent disturbance? I think as before you should be providing more reasoning behind these statements. What traits do cosmopolitan taxa possess that local communities?

L318: contribute to **increased** beta diversity across...

L330-334: You spend the majority of this paragraph highlighting the potential ways that agriculture might actually create more heterogeneous environments for soil microbes...I think your topic sentence for this paragraph should focus on the ways observations at different spatial scales can impact the interpretation of broad soil microbiome responses.

L352: these processes ***due to the breakdown of nutrient cycling plant-microbial symbioses under agricultural fertilization.***

L354-356: This statement is based solely on your analyses, with metagenomics which can amplify older (not functioning) genes. Perhaps you should clarify that other methods may reveal more realistic functional gene expression in the different land use scenarios, e.g. metatranscriptomics or qSIP, that correlate with the observed taxonomic shifts after agricultural conversion

L384: again you are basing this concept of functional redundancy on the analysis of DNA, which may not be correct for metabolically active functioning of the soil microbiome. I suggest you not focus on this concept in the concluding paragraph, particularly because there is no reason to suggest that agricultural land conversion (an environmentally detrimental thing across the globe) has little effect on microbial function... particularly when you actually found that there were significant decreases in N and P cycling. If you want to keep this concept, focus only on C metabolism/C cycling functional redundancy

L387: homogenization of ***the*** belowground microbiome

L389: crop performance? I don't know why this would be in here, perhaps just delete.

Response and actions taken with respect to the reviewer comments for:

NCOMMS-23-43840A

Title: Large-scale evidence of agriculture-induced taxonomic and functional
homogenization of belowground communities

Authors: Ziheng Peng ¹, Xun Qian ², Yu Liu ¹, Xiaomeng Li ¹, Hang Gao ¹, Yining An
¹, Jiejun Qi ¹, Lan Jiang ², Yiran Zhang ², Shi Chen ¹, Haibo Pan ¹, Beibei Chen ¹,
Chunling Liang ¹, Marcel G.A. van der Heijden ^{3,4}, Gehong Wei ^{1*}, Shuo Jiao ^{1*}

Reviewer #1 (Remarks to the Author):

I think the authors have satisfactorily addressed the concerns in the previous version of this manuscript. Overall, I am supportive of publication of this work.

Response: Thank you very much for your comments.

Minor comments:

The authors used both asterisks and P values in the same figure, such as Figure 1. Please try to use the method.

For the Abstract, can the authors further highlight how bacterial composition were altered, especially in Line 34-35. It is not clear how ‘41-45% of phylotypes’ altered? Are the abundances of these phylotypes decreased/lost or increased?

Response: We have adjusted the P value to be consistent in Figure 1. The abundance of 41-45% of phylotypes were altered (decreased or increased). We focused on the phylotypes increased in cropland, “We find that 41-45% of phylotypes are impacted by land conversion, with croplands enriched in Chloroflexi, Gemmatimonadota, Planctomycetota, Myxococcota and Latescibacterota”.

Reviewer #2 (Remarks to the Author):

Thanks for the comprehensive responses and corresponding amendments to the

manuscript, per my feedback and the valuable feedback of the other reviewers.

Glad you were able to clarify that enzyme activity wasn't simply inferred from the metagenomic data, and instead standard fluorometric assays measured enz max capacity.

Response: Thank you very much for your comments.

Reviewer #4 (Remarks to the Author):

Summary

This study analyzes continental (China) and global (meta-analysis) trends in land use conversion effects from natural ecosystems (e.g. wetland, grassland, forest) to agricultural production systems on soil bacterial communities and functions. They use a suite of methods, including amplicon sequencing, metagenomics and statistical analyses (e.g. FAPROTAX) to calculate function, and compare the changes across natural systems to determine if there are broad global consistencies. They show that agricultural conversion leads to more homogeneous bacterial communities (mostly relative abundance-based taxonomic similarity) both globally and continentally, and that these effects could have impacts on long-term soil bacterial community functions, particularly for N and P (less so for C metabolism functions).

General Comments

This is a very nice study with big implications given the scale of the analyses, and the importance of soil bacterial diversity and functionality undergoing massive land-use changes globally. Also, I should say that after reading previous reviewer comments, it appears that this manuscript is much more well organized and substantive than previous version/s, and generally conveys the scientific goals and questions well. That said, I think there are two things that need some attention, namely the word choice and general writing flow (grammar) and the Discussion section. I think some word choices are either not translating very well or, in my opinion, simply the wrong word used. For example, the authors repeatedly refer to 'agricultural intensification.' To me (and likely the reader), this implies that they are talking about different scales of agricultural management 'intensity', typically from sustainable-to-conventional (i.e. no-till, low

chemical use, mixed crop to tillage, high chemical input, monoculture), which would be incredibly interesting, but was not the focus of this paper, rather they looked at land conversion to some form of agriculture. Secondly, the Discussion still needs quite a bit of work, particularly, the authors need to 1) focus their discussion of other (prior) studies and provide substantive examples that link or contradict others' results, and 2) Align topic sentences with the entire paragraphs. While some of the topic sentences are interesting and well-worded, many times they don't actually fit the major point provided in the following sentences, therefore this requires some editing. Finally, I feel that while the figures (namely the trees in Figure 2) are all beautiful, there are too many (sub-figures) and some of the time, they do not convey enough (simply-interpretable) information for the reader. For example, it is almost impossible to determine what I should take away from the very large phylogenetic tree figures (Figs 2B,D,F). Therefore I suggest reviewing which figures you believe convey the most important information to the reader, and move the extras to supplemental files (see my more detailed comments regarding figures).

Response: Thank you very much for your comments. We have revised the text presentation and discussion sections based on the reviewers' suggestions. We replaced agricultural Intensification with agricultural conversion to better fit our theme. Moreover, we have revised the discussion and figures according to the comments, and moved the sub-figures in Figure 2 and Figure 4 to the supplemental files.

Reviewer #4 (Remarks on code availability):

The code found at: [https://github.com/Pong2021/Agricultural-impacts-on-soil-microbiome-function/blob/main/Beta diversity](https://github.com/Pong2021/Agricultural-impacts-on-soil-microbiome-function/blob/main/Beta%20diversity)

The code appears to be well-curated and accessible. While not an expert on this analysis, the 'estimation of ecological processes' seems to be properly constructed based on comparison of data to random null models using a beta NTI function.

Response: Yes, the estimation of ecological processes was performed according to Stegen et al. ¹ using β NTI. The aim of framework is to quantitatively estimate the degree to which spatial turnover in community composition is influenced by selection, drift

acting alone, dispersal limitation acting in concert with drift and homogenizing dispersal. The detailed information is showed in Methods.

Specific Comments

Figures

Figure 1 h + j - I do not understand how J would indicate such similarity while an effect size of KEGG betadiversity (based on Bray-Curtis?) and a PCoA of Bray-Curtis show such different results...?? This intuitively does not make sense given they appear to be using the same data and show very different results for grassland vs cropland KEGG function. For example, it would appear that the forest -> cropland functional comparison in Figure i would have stronger and likely significantly different functional results for the 'effect size', which we do not see in Figure h. This leaves me with questions rather than answers, please confirm data analysis/results are correct.

Response: Yes, this is the same set of data, but beta diversity analysis and PCoA analysis reveal the differences between farmland and natural ecosystems from different facets. PCoA analysis reveals differences in functional composition between farmland and natural systems, with the larger the composition difference, the farther apart the points are. Beta diversity analysis reveals community differences between sites within the same ecosystem, and then compares the two systems. The community differences between grassland sampling sites were larger than those in farmland (Fig. 1h), but the community differences between grassland and farmland were smaller (Fig. 1j).

Figure 2. While beautiful, it is very hard to discern the MOST important things to take away from these figures. The trees are beautiful, but it's very hard to gather the results. Perhaps keep A,C,E, (which have all the taxonomic and functional responses) and move B,D,F to supplement...? Also, in A,C,E please change the figure titles, as 'Agricultural effect on abundance in xx' does not make too much sense. Something like "Forest taxonomic & functional abundance relative to croplands". Also, in A,C,E please change the green descriptor (y axis) from 'Nutrient cycling' to 'Functionality' or "Function" as they are not all nutrient related responses...

Response: We have revised the text in Fig. 2, and moved the sub-figures (Fig. 2c-f) to the supplemental files and remained the Fig. 2a, b (between cropland and forest). We retained one of the three trees in main manuscript to show great variation across a broad range of phylotypes.

Figure 3. D - I suggest you keep blue as negative and red as positive to match other figures (e.g. Fig4 correlations and the SEM)

Response: Yes, we kept the consistent colors throughout the figures, but Fig. 4 and Fig. 3d showed different analysis and methods. Fig. 4 showed the correlations and links between environmental variables, and Fig. 3d showed the log₂-fold changes in croplands relative to natural ecosystems (forests, grasslands, and wetlands) using DESeq2 with the apeglm shrinkage algorithm.

Figure 4B - this is a very confusing figure to people that have not worked with this analysis, and to be direct, it tells me nothing unless you can create some sort of formatting that indicates what the differences in sites are, or if you can split the sites by altered community assembly processes on one side and unaltered assembly processes on the other to make it more visually easier to understand the point. Overall, 4C is much easier to understand and I suggest moving 4B to the supplement.

Response: We have moved Fig. 4B to the supplementary materials.

Fig 4E - You barely discuss this in the results or discussion section, and it is largely, in my opinion, an unimportant result. A consistently negative correlation between one principal coordinate axis of bacterial B-C dissimilarity and microbial functions in all ecosystem types is not terribly interesting, unless you present something you would like to discuss further, for example: more similar (less diverse) bacterial communities expressing more functions than less similar (more diverse) bacterial communities... which is slightly counterintuitive and could be interesting. But, given how many results you already have, I suggest moving 4E to the supplement.

Response: Since taxonomic composition (the first component of PCoA analysis) is

dimensionless, a negative correlation between taxonomic composition and soil functions simply indicates an association between them, not a negative correlation. In the same way, the relationship between taxonomic composition and soil functions also indicates an association, not a negative correlation. Thus, our results suggest that taxonomic composition is closely related to soil functions based metagenomic data sites and all sites (Fig. 4e).

Manuscript

L24: can just say aboveground

Response: Done.

L33: “loss of soil bacteria with restricted ranges (endemic)” is a very specific conclusion regarding dispersal capacity, niche breadth, etc... I think this terminology may have to be proven more definitively to say this. Additionally, it does not make too much sense that the ‘loss’ of taxa with ‘restricted ranges’ would lead to homogenization. I suggest rewriting: “Our results demonstrate large-scale bacterial taxonomic and functional homogenization upon conversion to agriculture, mainly driven by the loss of soil bacteria.”

Response: Done, revised as suggested.

L39: I would suggest adding: “reduction in functional diversity provided by soil bacteria in cropland soil.”

Response: Done.

L31-36: I suggest moving sentences from L33-36 (Agricultural...Latescibacterota) above the previous sentence (Our results...)

Response: Done, we have moved these two sentences.

L49: revise: have focused on local species diversity ...

Response: Done.

L50-51: ...are relevant to highlight the loss of global biodiversity...

Response: Done.

L84: isn't it really agricultural conversion rather than agricultural intensification here ??

Response: Done. We have replaced “agricultural intensification” with “agricultural conversion” to better fit our theme in the revised MS.

L86: i would suggest adding: ...biodiversity and associated ecosystem functions.

Response: Done.

L91: most strongly impacted?

Response: Yes, done.

L101: For the second question,

Response: Done.

L106: land-use change, and demonstrate ... scales is a consequence of land-use intensification through agricultural conversion.

Response: Done.

L115: cropland soils are more similar than paired natural ecosystem soils.

Response: Done.

L125: loss of habitat-specific taxa

Response: Done.

L127-132: This section is a stretch for me. I would not consider there being slightly more samples from croplands than the original habitat type that harbor shared taxa (ubiquitous) being a range expansion. In fact, because in many cases these samples

would be from land conversions e.g. from forest to agriculture, therefore the ‘range’ is just a different habitat type. Secondly, on L131-132 ‘loss of habitat-specific endemics in croplands’ is a little confusing as well... especially when considering SFig 6A (>50% of taxa are unchanged when converted to ag, and fairly similar proportions that are ‘enriched’ or ‘deleted’). Croplands surely also have their own ‘unique’ taxa that are not shown in the figures, so yes, the community changes (richness and beta diversity) and the taxa that are unique (‘endemic’) to e.g. grasslands are lost, but 1) there are many more shared taxa than unique taxa and 2) there are also habitat-specific endemics in croplands.

My issue with this phrasing trying to connect bacteria taxa to endemism or range expansion is that it assumes a lot of functional/physiological specificity for bacteria based on sequencing data from soil. It’s not that I don’t believe it might be true, but it feels like an unnecessary logical leap to concepts that are almost untestable in reality (thousands of taxa that can’t be cultured). So my suggestion is to re-write and clarify this section or to simply present the results without suggesting range expansion or loss of ‘endemics’ (which I don’t think is a term that should be used for taxa unique to particular soil sample’s habitat type anyway.)

In my opinion, homogenization implies lots of bad things, including the simplification/loss of really important functional capacities (associated with diversity), and so these specific definitions of ‘biotic homogenization’ you have used may be unnecessary and this section can be stated more simply. I suggest this paper for some other context regarding homogenization etc: Kortz &Magurran 2019 - <https://royalsocietypublishing.org/doi/10.1098/rsbl.2019.0133>

Response: Thank you for your comments. Biotic homogenization can occur through the loss of taxa with restricted geographic ranges and/or rare taxa, and the increase of taxa with broad geographic ranges and/or abundant taxa, and the increase in the ranges of existing species (e.g., through the removal of dispersal barriers). In our study, we attempted to explore the underlying mechanisms of biotic homogenization caused by agricultural conversion. We clarify this section to simply present the results, “Moreover, we found that the phylotypes, that present in both croplands and natural ecosystems,

were found in significantly more samples of croplands than in natural ecosystems (wilcoxon test: $p < 0.001$; Supplementary Fig. 4a), indicating an increase in the geographic ranges of existing taxa in croplands. The phylotypes unique to natural ecosystems occurred in significantly fewer samples than other shared phylotypes that present in both croplands and natural ecosystems (wilcoxon test: $p < 0.001$; Supplementary Fig. 4b), implying a possible loss of these habitat-specific taxa after agricultural conversion”.

L176: observed in the metagenomic dataset

Response: Done.

L177-178: what is meant by ‘categories’? Functional classifications from what analysis metagenomics mapped to KEGG??

Edit: functional categories found in croplands exhibited consistent changes in direction when compared to the three natural ecosystems after aggregating all metagenomic? functional category count data? from KEGG/FAPROTAX? above level three (I’m not sure what the data are here? specify what data and what analysis this is referring to).

Response: Done. Metagenomic data with COG functional annotations were used here. We have added the COG database in this sentence. In previous sentence, we pointed out that this section of the results comes from the metagenomic data, “in the metagenomic dataset (Fig. 3 and Supplementary Fig. 7)”.

L181: what are degradative genes (carbon-degrading?), please be more specific.

Response: Yes, carbon-degrading genes. We have added it in the revised MS.

L182: when you say ‘agricultural intensification’ I think you should be saying instead agricultural conversion. As you do not have treatment/s or treatment gradients for these analyses (for example different levels of fertilization, plowing, pesticide application, etc which would be associated with ag intensification).

Response: Yes, we have replaced “agricultural intensification” with “agricultural

conversion”.

L188: make this a new paragraph at Agriculture significantly...

Response: We started a new paragraph in the revised MS.

L189: a number of functionally important N cycling, P utilization, and sulfur metabolism genes

Response: Done.

L193: application of fertilizers and/or the loss of leguminous plant taxa found in natural ecosystems. (important to also consider the loss of important plant taxa in ag that affect microbial functions such as nifH as well)

Response: Yes, we have added this important information in the revised MS.

L206-206: I like when you introduce the ecological importance/role of these gene/trait responses, but I think you should elaborate on this resuscitation-promoting gene here. Use a citation that highlights that this means and why this makes sense in croplands versus e.g. a forest

Response: We have added the relevant explanation and citation, “We also observed that resuscitation-promoting gene was increased in cropland (Supplementary Fig. 8), which are associated with long-term persistence of viable bacterial populations ², indicating that the resuscitation after disturbance can allow for the proliferation of dormant taxa and accelerate increases in species richness ³” in the revised MS.

L209: I think you should specify how community assembly was calculated when presenting this result. For example: ...dominated microbial community assembly (as calculated using...analysis) in croplands...

Response: Done.

L211: At the same time, agriculture, acting as... (delete ‘impacts’)

Response: Done.

L211: filter, continues

Response: Done.

L213: again, I'm not sure if 'intensification' is the word you want here. This point regarding agricultural management leading to more homogenous abiotic and biotic conditions is nice, but agricultural management does also imply a lot of seasonal disturbances (e.g. fertilization, chemical additions, OM additions, plowing, etc.), so while they may be consistent disturbances I think you want to avoid the word 'intensification' as it suggests management practices will be progressively more disruptive, which then runs counter to the concept of abiotic/biotic homogenization for the soil bacterial tax. Please keep an eye on this word throughout the manuscript and be sure you actually mean intensification and not something like conventional management practices (such as plowing, pesticide and fungicide applications, etc.)

Response: We avoid the word "intensification" and instead use "agricultural conversion" in the revised MS.

L226: composition or abundance??

Response: Composition is better because microbial taxonomic composition (16S) and functional (KEGG) composition was represented by the principal coordinate analyses 1, the first component of PCoA analysis.

L238: what is meant by a 'coupled relationship'? Just be more explicit with what you mean here, because as written it just seems like the metagenomic data and the enzyme data did not reveal the same results (actually suggesting a potential problem with methods being able to consistently resolve microbial functioning in soil).

Response: There is no relationship between microbial functional composition and soil enzyme functions. We have revised this sentence, "we did not observe the relationship between microbial functional composition and soil enzyme functions" in the revised

MS.

L255: different agricultural practices?? I didn't see results differentiating the type of agricultural management/cropping system, etc?

Response: In the previous revision, experiments with different agricultural practices were removed. We have deleted this phrase.

L256: of taxonomic and, to a lesser degree, functional homogenization

Response: Done.

L257: delete 'landscapes'

Response: Done.

L258: taxonomic variation

Response: Done.

L259: significantly lower in croplands than in grasslands, ...

Response: Done.

L263-266: I don't really understand why the positive effects of agriculture on aboveground diversity (richness?) are being discussed here? Including the relative increases in 'exotic' species?? This is just kind of strange, and doesn't fit. I suggest deleting this part and getting from a better topic sentence to the relationships observed where alpha metrics can increase at the expense of betadiv.

Response: Done. We have deleted this section in the revised MS.

L270: ... animal groups both above- and below-ground

Response: Done.

L277: I think it is important to remind the reader that the functional results (though significant in some cases) were not as strong as the taxonomic results. This is still a

very important finding, but you should simply state something like: “Although taxonomic homogenization in cropland versus natural ecosystems was stronger and more significant in many cases, we observed very important microbial functional shifts under croplands, including functional homogenization.”

Response: Done. We have replaced “Moreover, biotic homogenization not only led to a decrease in community dissimilarity (i.e., taxonomic homogenization) in our study but also to a reduction in spatial variation of functional traits, a process called functional homogenization” with “Although taxonomic homogenization in cropland versus natural ecosystems was stronger and more significant in many cases, we observed very important microbial functional shifts under croplands, including functional homogenization” in the revised MS.

L283-286: I’m not really sure what the purpose of this statement is, and also it doesn’t make much sense to me. How would massive amounts of fertilizer inputs “likely not only to be the cause for reduced functional diversity.” ??? Perhaps this is just a grammatical error, or you are trying to make a different point, but you should edit this sentence. You then go on to state, “external inputs may also compensate for reduced functional diversity and functioning.” As written, it appears you are suggesting that we can just use agricultural chemicals (that are often bad for the environment) to offset the losses of microbial functioning in soil? I think this statement should be either removed or revised to be more explicit about your meaning

Response: We have removed this sentence in the revised MS.

L288: I think more detail is needed for these two studies you cite, for example taxonomic homogenization was identified for EU grasslands compared to agricultural intensification? Or was there just some sort of management of the grassland that homogenized taxa/functions? Readers would not understand exactly the connection between your analysis and the two papers cited.

Response: We have revised this sentence for clarity, “Our findings extend taxonomic-level results in Amazonian Forest ⁴ and European grasslands ⁵ that focus on the impact

of agricultural management on belowground taxonomic homogenization in local-scale, to the large-scale functional homogenization”.

L291: what are the major implications? That we should stop converting land to agriculture? If so, say that. If you mean some more complex land use management strategy/national regulations with enforcement plans, suggest that. As written, this is kind of an empty statement.

Response: Yes, we have written this sentence for clarity, “our study provides a comprehensive insight that agricultural land-use change cause biotic homogenization in taxonomic and functional composition, and suggests halting reclamation and developing ecological restoration for cropland to conserve landscape-scale biodiversity and ecosystem service provision ^{6,7}”.

L298: I don’t know what this means??: cause some general effects on soil communities ?? Rewrite this

Response: We have removed “some general effects” and rewritten this sentence, “This highlights the selective effects of agricultural conversion, which could cause pressure and force on soil communities from natural ecosystems. For example, the destruction of soil structure ...”.

L301: result in the trait-based filtering out of certain species, leading to...

Response: Done.

L303: what do you mean by habitat filtering?? Provide a brief example to allow the reader to understand how abundant colonizer species would benefit and ‘win’ compared to a specialist.

Response: We have using intensive management instead of habitat filtering. We have added the relevant explanation before the sentence, “Moreover, geographic range size is a key determinant of species’ extinction risk, and rare species therefore are vulnerable to land use change and are at greater risk of extinction ⁸”, and after the sentence, “Land-

use change is proposed to affect turnover in community composition via its effect on stress tolerance, resource acquisition, and dispersal ability. Stronger stress-tolerant, broader resource-flexibility cosmopolitan species with unlimited dispersal capacity are more stable to land-use change because of increasing adaptive potential and/or extensive ability to exploit soil resource availability ^{9, 10}” in the revised MS.

L305-306: similarly, why wouldn't there just be new specialist taxa in the new agriculture habitat? There would be some 'loser' specialists and some 'winner' specialists according to this logic

Response: Yes, maybe there are new specialist taxa in the new agriculture habitat. Here we focus on the effect of geographic range size, and disproportionate impact of human land use on large- and small-ranged species. Compared the small-ranged species, large-ranged species are more likely to be present in human land uses, leading to a homogenization of assemblage composition across space.

L308-309: why would cosmopolitan communities be able to outcompete locally distinct communities after frequent disturbance? I think as before you should be providing more reasoning behind these statements. What traits do cosmopolitan taxa possess that local communities?

Response: We have added the relevant trait information, “Land-use change is proposed to affect turnover in community composition via its effect on stress tolerance, resource acquisition, and dispersal ability. Stronger stress-tolerant, broader resource-flexibility cosmopolitan species with unlimited dispersal capacity are more stable to land-use change because of increasing adaptive potential and/or extensive ability to exploit soil resource availability ^{9, 10}” in the revised MS.

L318: contribute to increased beta diversity across...

Response: Done.

L330-334: You spend the majority of this paragraph highlighting the potential ways that agriculture might actually create more heterogeneous environments for soil microbes...I think your topic sentence for this paragraph should focus on the ways observations at different spatial scales can impact the interpretation of broad soil microbiome responses.

Response: Yes, This paragraph and the previous paragraph discuss two facets of biotic homogenization caused by agriculture, the invasion and extinction of species and heterogeneity of landscapes, respectively. At the end of this paragraph, we showed that different spatial scales can impact the interpretation of broad soil microbiome responses, “more attention needs to be paid to the biotic homogenization caused by agricultural conversion of the soil microbiome at various spatial scales”.

L352: these processes due to the breakdown of nutrient cycling plant-microbial symbioses under agricultural fertilization.

Response: Thank you for your addition. We have added the phrase in the revised MS.

L354-356: This statement is based solely on your analyses, with metagenomics which can amplify older (not functioning) genes. Perhaps you should clarify that other methods may reveal more realistic functional gene expression in the different land use scenarios, e.g. metatranscriptomics or qSIP, that correlate with the observed taxonomic shifts after agricultural conversion

Response: We have removed this sentence and added the advanced roads for future research, “More realistic functional gene expression study the functional divergences, redundancies, and complementarities in the different land use scenarios, e.g. metatranscriptomics ¹¹ or quantitative stable-isotope probing (qSIP) ¹², that correlate with the observed taxonomic shifts after agricultural conversion, needs to be further revealed in the future” in the revised MS.

L384: again you are basing this concept of functional redundancy on the analysis of DNA, which may not be correct for metabolically active functioning of the soil microbiome. I suggest you not focus on this concept in the concluding paragraph, particularly because there is no reason to suggest that agricultural land conversion (an environmentally detrimental thing across the globe) has little effect on microbial function... particularly when you actually found that there were significant decreases in N and P cycling. If you want to keep this concept, focus only on C metabolism/C cycling functional redundancy

Response: We have pointed out functional redundancy of C metabolism in the revised MS.

L387: homogenization of the belowground microbiome

Response: Done.

L389: crop performance? I don't know why this would be in here, perhaps just delete.

Response: We have deleted it in the revised MS.

Reference

1. Stegen JC, *et al.* Quantifying community assembly processes and identifying features that impose them. *ISME J* **7**, 2069-2079 (2013).
2. Kana BD, Mizrahi V. Resuscitation-promoting factors as lytic enzymes for bacterial growth and signaling. *FEMS Immunol Med Microbiol* **58**, 39-50 (2010).
3. Kearns PJ, Shade A. Trait-based patterns of microbial dynamics in dormancy potential and heterotrophic strategy: case studies of resource-based and post-press succession. *ISME J* **12**, 2575-2581 (2018).
4. Rodrigues JLM, *et al.* Conversion of the Amazon rainforest to agriculture results in biotic homogenization of soil bacterial communities. *Proceedings of the National Academy of Sciences* **110**, 988-993 (2013).
5. Gossner MM, *et al.* Land-use intensification causes multitrophic homogenization of grassland communities. *Nature* **540**, 266-269 (2016).
6. van der Plas F, *et al.* Biotic homogenization can decrease landscape-scale forest multifunctionality. *Proc Natl Acad Sci USA* **113**, 3557-3562 (2016).
7. Le Provost G, *et al.* The supply of multiple ecosystem services requires biodiversity across spatial scales. *Nat Ecol Evol*, (2022).
8. Banerjee S, *et al.* Biotic homogenization, lower soil fungal diversity and fewer rare taxa in arable soils across Europe. *Nat Commun* **15**, 327 (2024).
9. Bell TH, Bell T. Many roads to bacterial generalism. *FEMS Microbiology Ecology* **97**, fiae240 (2021).
10. Mueller RC, Rodrigues JLM, Nüsslein K, Bohannan BJM. Land use change in the Amazon rain forest favours generalist fungi. *Funct Ecol* **30**, 1845-1853 (2016).
11. Peng J, Zhou X, Rensing C, Liesack W, Zhu Y-G. Soil microbial ecology through the lens of metatranscriptomics. *Soil Ecology Letters* **6**, 230217 (2023).
12. Hungate Bruce A, *et al.* Quantitative Microbial Ecology through Stable Isotope Probing. *Applied and Environmental Microbiology* **81**, 7570-7581 (2015).